



# Mapping the drivers of uncertainty in atmospheric selenium deposition with global sensitivity analysis

Aryeh Feinberg[1,2,3], Moustapha Maliki[4], Andrea Stenke[1], Bruno Sudret[4], Thomas Peter[1], and Lenny H. E. Winkel[2,3]

[1]Institute for Atmospheric and Climate Science, ETH Zurich, Zurich, Switzerland
[2]Institute of Biogeochemistry and Pollutant Dynamics, ETH Zurich, Zurich, Switzerland
[3]Eawag, Swiss Federal Institute of Aquatic Science and Technology, Dübendorf, Switzerland
[4]Chair of Risk, Safety and Uncertainty Quantification, ETH Zurich, Zurich, Switzerland

**Correspondence:** Aryeh Feinberg (aryeh.feinberg@env.ethz.ch)

**Abstract.** An estimated 0.5–1 billion people globally have inadequate intakes of selenium (Se), due to a lack of bioavailable Se in agricultural soils. Deposition from the atmosphere, especially through precipitation, is an important source of Se to soils. However, very little is known about the atmospheric cycling of Se. It has therefore been difficult to predict how far Se travels in the atmosphere and where it deposits. To answer these questions, we have built the first global atmospheric Se model by implementing Se chemistry into an aerosol–chemistry–climate model, SOCOL-AER. In the model, we include information from the literature about the emissions, speciation, and chemical transformation of atmospheric Se. Natural processes and anthropogenic activities emit volatile Se compounds, which oxidize quickly and partition to the particulate phase. Our model tracks the transport and deposition of Se in 7 gas-phase species and 41 aerosol tracers. However, there are large uncertainties associated with many of the model's input parameters. In order to identify which model uncertainties are the most important for understanding the atmospheric Se cycle, we conducted a global sensitivity analysis with 34 input parameters related to Se chemistry, Se emissions, and the interaction of Se with aerosols. In the first bottom-up estimate of its kind, we have calculated a median global atmospheric lifetime of 4.4 d (days), ranging from 2.9–6.4 d ($2^{nd}$–$98^{th}$ percentile) given the uncertainties of the input parameters. The uncertainty in the Se lifetime is mainly driven by the uncertainty in the carbonyl selenide (OCSe) oxidation rate and the lack of tropospheric aerosol species other than sulfate aerosols in SOCOL-AER. In contrast to uncertainties in Se lifetime, the uncertainty in deposition flux maps are governed by Se emission factors, with all four Se sources (volcanic, marine biosphere, terrestrial biosphere, and anthropogenic emissions) contributing equally to the uncertainty in deposition over agricultural areas. We evaluated the simulated Se wet deposition fluxes from SOCOL-AER with a compiled database of rainwater Se measurements, since wet deposition contributes around 80% of total Se deposition. Despite difficulties in comparing a global, coarse resolution model with local measurements from a range of time periods, past Se wet deposition measurements are within the range of the model's $2^{nd}$–$98^{th}$ percentile at 79% of background sites. This agreement validates the application of the SOCOL-AER model to identifying regions which are at risk of low atmospheric Se inputs. In order to constrain the uncertainty in Se deposition fluxes over agricultural soils we should prioritize field campaigns measuring Se emissions, rather than laboratory measurements of Se rate constants.



# 1 Introduction

Selenium (Se) is an essential dietary trace element for humans and animals, with the recommended intake ranging between 30 and 900 $\mu g\,d^{-1}$ (Fairweather-Tait et al., 2011). The amount of Se in crops depends on the amount of bio-available Se in the soils where the crops are grown (Winkel et al., 2015). Levels of Se in soils, as well as Se dietary intakes, vary strongly around

the world (Jones et al., 2017). Selenium deficiency is considered a more widespread issue than Se toxicity, as around 0.5 to 1 billion people are estimated to have insufficient Se intakes (Fordyce, 2013).

Atmospheric deposition is an important source of Se to soils. In several regions, Se concentrations in soils were found to correlate with precipitation (Låg and Steinnes, 1978; Blazina et al., 2014). As well, several studies have attributed an increase in soil Se concentrations to regional anthropogenic Se emissions to the atmosphere (Haygarth et al., 1993; Dinh et al., 2018),

suggesting a link between atmospheric Se inputs and soil concentrations. However, apart from some budget studies in the 1980's (Ross, 1985; Mosher and Duce, 1987), there has been a lack of research into global-scale atmospheric cycling of Se and the spatial patterns of Se deposition.

## 1.1 Atmospheric Se cycle

Since Se and sulfur (S) are in the same group on the periodic table, they share chemical properties and their biogeochemical

cycles are similar. Like S, Se is emitted to the atmosphere by both natural and anthropogenic sources, with the total annual emissions estimated between 13 and 19 $Gg\,Se\,yr^{-1}$. Natural sources of atmospheric Se include volatilization by the marine and terrestrial biospheres, volcanic degassing and eruptions, and minor contributions from sea salt and mineral dust. Anthropogenic Se is emitted during coal and oil combustion, metal smelting, and biomass burning. However, very few in-situ measurements of Se emissions fluxes and speciation exist. Once in the atmosphere, volatile Se species are oxidized, eventually forming species

like elemental Se and $SeO_2$ (Wen and Carignan, 2007). These oxidized species are expected to partition to the particulate phase; previous measurements have found that 75–95% of Se is in particulates (Mosher and Duce, 1983, 1987). The fate of atmospheric Se is dry and wet deposition, with wet deposition accounting for an estimated 80% of total deposition globally (Wen and Carignan, 2007).

Atmospheric chemistry modeling studies have been applied to other trace elements to predict atmospheric lifetimes and

spatial patterns of deposition. For example, atmospheric mercury models were developed more than two decades ago (Petersen and Munthe, 1995), and now there are around 16 global and regional atmospheric mercury models (Ariya et al., 2015). A recent mechanistic modeling paper has advanced the understanding of atmospheric arsenic cycling (Wai et al., 2016). To our knowledge, Se chemistry has never previously been included in an atmospheric chemistry–climate model (CCM), and thus many questions surrounding atmospheric Se transport remain unanswered. For example, the atmospheric lifetime of Se and

thus the scales at which it can be transported (local, regional, hemispheric, global) are not well constrained.



## 1.2 Global sensitivity analysis

Since the atmospheric Se cycle has been investigated only by a limited number of studies, it is essential that we consider the relevant parametric uncertainties when building an atmospheric Se model. The reaction rate coefficients of Se compounds have either only been measured by one laboratory study, or no laboratory measurement exists and these rate coefficients need to be

estimated. Selenium emission fluxes from certain sources have been measured, however it remains difficult to extrapolate these measurements to global fluxes due to the high degree of spatial and temporal variability. Atmospheric Se modeling can only be considered trustworthy when combined with full accounting of input parameter uncertainties and their propagation through the model. Through "global sensitivity analysis" (Saltelli et al., 2008) we can identify which input uncertainties are the most important for the uncertainty in the model output. A sensitivity analysis is called *global* when the sensitivity is evaluated over

the entire input parameter space, as opposed to *local* methods that test sensitivity only at a certain reference point in the space (i.e., based on the gradient of the output at this reference point). Sensitivity analysis provides a framework to prioritize which model inputs should be further constrained in order to reduce the uncertainty in the model output.

Until recently, most sensitivity analyses of atmospheric chemistry models consisted of local methods, principally the one-at-a-time approach (OAT). In OAT, the model is initially run with a set of default parameters to yield a "reference" simulation.

Multiple sensitivity simulations are then conducted, so that for each simulation one parameter is perturbed from the reference set at a time. The influence of these perturbations on the model output of interest would then be analyzed. However, this approach may be flawed because it only considers the first order response of the model to each parameter, ignoring interactions that might exist between parameters (Saltelli et al., 2008; Lee et al., 2011). As well, the uncertainty ranges of the input parameters are rarely quantified and reported; much of the possible parameter space often remains unexplored. Global methods

such as variance-based sensitivity analysis allow the uncertainty in model output to be apportioned to each input variable. Sobol' indices, which represent the fraction of model variance that one input variable explains, provide a ranking system for the importance of input variables (Sobol, 1993). The benefits of global sensitivity analysis include: 1) identifying the most influential input variables, i.e., the ones that should be further constrained to yield the biggest reduction in model uncertainty; 2) identifying input variables that do not play any role in the model output variance; this could represent a route to simplify

the model, since the process involving these input variables can be neglected; 3) understanding the behavior of the model, for example how the output depends on interactions between input variables; 4) identifying possible model bugs or discontinuities, since the model will be tested with a wide range of input parameter values (Saltelli et al., 2000; Ryan et al., 2018).

There are several recent examples of atmospheric chemistry studies that included a global sensitivity analysis. The sensitivity of cloud condensation nuclei number density to input parameters in an aerosol model was investigated at the local (geographic)

scale (Lee et al., 2011) and at the global scale (Lee et al., 2012, 2013), revealing regional differences in parameter rankings. Revell et al. (2018) investigated the sensitivity of the tropospheric ozone columns to emission and chemical parameters, to identify which processes are responsible for the bias in modeled tropospheric ozone. Marshall et al. (2019) employed global sensitivity analysis methods to identify how radiative forcing responds to volcanic emission parameters. In these examples, surrogate modeling techniques (also known as emulation) were employed to replace a process-oriented, computationally ex-





pensive model with an approximative statistical model. The statistical model has the advantage that it is quicker to evaluate; therefore, it can be used to calculate the model output throughout the parametric space (Lee et al., 2011). The examples given above all used Gaussian process emulation, however other surrogate modeling techniques exist, including polynomial chaos expansions (PCE) (Ghanem and Spanos, 2003). The PCE approach is well-suited to sensitivity analysis, since the Sobol' sen-
sitivity indices can be extracted analytically from the constructed PCE, with no need to evaluate the surrogate model through Monte Carlo sampling (Sudret, 2008). This can greatly reduce the computational time required to conduct the sensitivity analysis, especially when one is interested in conducting a separate sensitivity analysis for many model grid boxes.

### 1.3 Outline

This study focuses on the construction of the first global atmospheric Se model and the insights that this model reveals into
which Se cycle uncertainties would be the most important to constrain. Section 2 describes the SOCOL-AER model and the implementation of Se chemistry in the SOCOL-AER model. The SOCOL-AER model is a suitable tool to model the Se cycle, since it successfully describes the major properties of the atmospheric S cycle (Sheng et al., 2015; Feinberg et al., 2019b). The statistical methods that we use to conduct the sensitivity analysis are discussed in Sect. 3. Section 4 details the methods used to compile a database of measured wet Se deposition fluxes, which we use to evaluate the model. The results of the
sensitivity analyses are presented in Sect. 5.1 for the atmospheric Se lifetime and Sect. 5.2 for the Se deposition patterns. Section 5.3 illustrates the comparison between the compiled Se deposition measurements and simulated results. A discussion of both sensitivity analyses follows in Sect. 6, and the paper is concluded in Sect. 7.

## 2 Model Description

### 2.1 SOCOL-AER

SOCOL-AERv2 is a global CCM that includes a coupled sulfate aerosol microphysical scheme (Sheng et al., 2015; Feinberg et al., 2019b). The base CCM, SOCOLv3 (Stenke et al., 2013), is a combination of the general circulation model ECHAM5 (Roeckner et al., 2003) and the chemical model MEZON (Egorova et al., 2003). The MEZON submodel comprises a comprehensive atmospheric chemistry scheme, with 89 gas-phase chemical species, 60 photolysis reactions, 239 gas-phase reactions, and 16 heterogeneous reactions. Chemical tracers are advected in the model using the Lin and Rood (1996) semi-Lagrangian
method. Photolysis rates are calculated using a look-up table approach based on the simulated overhead ozone and oxygen columns. The MEZON model solves the system of differential equations representing chemical reactions with a Newton-Raphson iterative method for short-lived chemical species and an Euler method for long-lived species.

The sulfate aerosol model AER (Weisenstein et al., 1997) was first coupled to SOCOL by Sheng et al. (2015). SOCOL-AER includes gas-phase S chemistry and 40 sulfate aerosol tracers, ranging in dry radius size from 0.39 nm to 3.2 μm. SOCOL-
AER simulates microphysical processes that affect the aerosol size distribution, including binary homogeneous nucleation, condensation of $H_2SO_4$ and $H_2O$, coagulation, evaporation, and sedimentation. SOCOL-AER was extended in Feinberg et al.





(2019b) to include interactive wet and dry deposition schemes. The wet deposition scheme, based on Tost et al. (2006), calculates scavenging of gas-phase species depending on their Henry's law coefficients and aerosol species depending on the particle diameter. The wet removal of tracers is coupled to the grid cell simulated properties of clouds and precipitation. The dry deposition scheme is based on Kerkweg et al. (2006, 2009), which uses the surface resistance approach of Wesely (1989).

In addition to surface type and meteorology, the calculated dry deposition velocities depend on reactivity and solubility for gas-phase compounds and size for aerosol species. SOCOL-AER uses an operator splitting approach, wherein the model time step is 2 hours for chemistry and radiation and 15 minutes for dynamics and deposition. Aerosol microphysical routines use a sub-time step of 6 minutes.

    For the simulations in this study we use boundary conditions for the year 2000. Sea ice coverage and sea surface temperatures

are prescribed from the Hadley Centre dataset (Rayner et al., 2003). Year 2000 concentrations of the most relevant greenhouse gases ($CO_2$, $CH_4$, and $N_2O$) derive from NOAA observations (Eyring et al., 2008). Anthropogenic CO and $NO_x$ emissions are based on the RETRO dataset (Schultz and Rast, 2007), while natural emissions are taken from Horowitz et al. (2003). Sulfur dioxide ($SO_2$) emissions from anthropogenic sources follow the year 2000 inventory from Lamarque et al. (2010) and Smith et al. (2011). Volcanic degassing $SO_2$ emissions are assigned to surface grid boxes where volcanoes are located (Andres and

Kasgnoc, 1998; Dentener et al., 2006). Atmospheric emissions of dimethyl sulfide (DMS) are calculated using a wind-based parametrization (Nightingale et al., 2000) and a marine DMS climatology (Lana et al., 2011). To represent mean conditions for photolysis, the look-up table for photolysis rates is averaged over two solar cycles (1977–1998).

## 2.2   Implementing Se emissions and chemistry in SOCOL-AER

### 2.2.1   Selenium species overview

We included the Se cycle in SOCOL-AER (Fig. 1) based on the existing literature on atmospheric Se (Ross, 1985; Wen and Carignan, 2007). Seven Se gas-phase tracers have been added to SOCOL-AER (Table 1): carbonyl selenide (OCSe), thiocarbonyl selenide (CSSe), carbon diselenide ($CSe_2$), dimethyl selenide (DMSe), hydrogen selenide ($H_2Se$), oxidized inorganic Se (OX_Se_IN), and oxidized organic Se (OX_Se_OR). The oxidized inorganic Se tracer represents species such as elemental Se, selenium dioxide ($SeO_2$), selenous acid ($H_2SeO_3$), and selenic acid ($H_2SeO_4$). Very little is known about the kinetics of inter-

conversion between the oxidized inorganic Se species (Wen and Carignan, 2007), and therefore in our model they are treated as one tracer. However, these species all have very low vapor pressures under atmospheric conditions (Rumble, 2017) and likely partition to the particulate phase. Oxidized organic Se species include dimethyl selenoxide (DMSeO) and methylseleninic acid ($CH_3SeO_2H$), which form after oxidation of DMSe (Atkinson et al., 1990; Rael et al., 1996). Similar to the oxidized inorganic Se compounds, oxidized organic Se species also partition to the particulate phase due to their low volatilities (Rael et al., 1996).

## 2.2.2   Selenium emissions

To determine which Se compounds are emitted by the different sources, we have reviewed studies that investigated the speciation of Se emissions. Thermodynamic modeling and in situ measurements of combustion exhaust gases have detected the





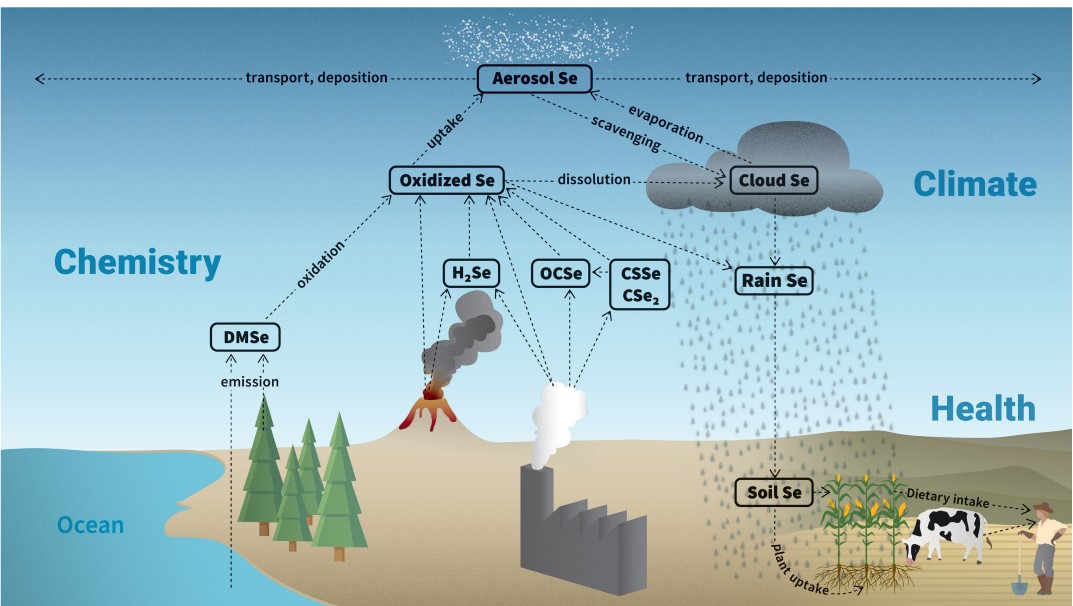

**Figure 1.** Scheme of the atmospheric Se cycle in SOCOL-AER, based on information in Ross (1985); Wen and Carignan (2007). For simplicity the two oxidized Se tracers in SOCOL-AER are represented with a single box. The impact on agriculture and human health is also shown, since it motivates the study of atmospheric Se.

following Se species in anthropogenic emissions: oxidized inorganic Se, $H_2Se$, OCSe, $CSe_2$, and CSSe (Yan et al., 2000, 2004; Pavageau et al., 2002, 2004). Oxidized inorganic Se species and minor amounts of $H_2Se$ are expected to be emitted from volcanic degassing (Symonds and Reed, 1993; Wen and Carignan, 2007). A variety of methylated Se species have been observed from biogenic emissions, including DMSe, dimethyl diselenide (DMDSe), dimethyl selenylsulfide (DMSeS), and methane se-

lenol (MeSeH) (Amouroux and Donard, 1996, 1997; Amouroux et al., 2001; Wen and Carignan, 2007). Since DMSe is usually the dominant emitted compound, and little is known about the oxidation kinetics of the other methylated species, DMSe is the only species emitted by marine and terrestrial biogenic emissions in SOCOL-AER.

Atmospheric S emissions have been measured more extensively than Se emissions, so we scale inventories of S emissions to yield the spatial distribution of emitted Se. For the sensitivity analysis we assume that the Se emissions have the same

distribution as S emissions, and we focus on the uncertainties in the global scaling factors for each source. The range in scaling factors will be discussed in Sect. 3.1.4. The spatial distribution of anthropogenic and biomass burning Se emissions comes from the $SO_2$ inventory for the year 2000 (Lamarque et al., 2010; Smith et al., 2011). For sea surface DMSe concentrations, we scale the DMS climatology calculated by Lana et al. (2011), and calculate DMSe emissions using a wind-driven parametrization (Nightingale et al., 2000). The locations and strength of background degassing volcanic emissions are taken from the GEIA

inventory (Andres and Kasgnoc, 1998; Dentener et al., 2006). Since little is known about both terrestrial biogenic Se emissions and terrestrial S emissions (Pham et al., 1995), we assume that terrestrial Se is emitted in all land surface grid boxes, excluding glaciated locations.





**Table 1.** Description of Se tracers included in SOCOL-AER

| Abbreviation | Name | Sources | Sinks |
|---|---|---|---|
| OCSe | Carbonyl selenide | Anthropogenic emissions, chemical production | Chemical loss |
| CSSe | Thiocarbonyl selenide | Anthropogenic emissions | Chemical loss |
| $CSe_2$ | Carbon diselenide | Anthropogenic emissions | Chemical loss |
| DMSe | Dimethyl selenide | Marine and terrestrial emissions | Chemical loss |
| $H_2Se$ | Hydrogen selenide | Anthropogenic and volcanic emissions | Chemical loss |
| OX_Se_IN | Oxidized inorganic Se | Anthropogenic and volcanic emissions, chemical production | Dry and wet deposition |
| OX_Se_OR | Oxidized organic Se | Anthropogenic and volcanic emissions, chemical production | Dry and wet deposition |
| - | Se in S aerosol (40 tracers) | Uptake of gas-phase oxidized Se | Dry and wet deposition |
| - | Se in dummy aerosol | Uptake of gas-phase oxidized Se | Dry and wet deposition |

### 2.2.3 Chemistry of Se species

We conducted a literature review to develop the model's chemical scheme of the Se cycle. Reactions of Se species that have been measured by laboratory studies are compiled in Table 2. We neglect any temperature dependency in the Se reaction rates, since the Se reactions have only been studied at around 298 K. For all compiled reactions, atmospheric Se compounds react

much quicker than the analogous S compounds, due to the stronger bonds that S forms with carbon and hydrogen than Se (Rumble, 2017). In addition, there are reactions that are known to occur for the analogous S compound, but have never been studied for the Se compound (OCSe + OH, CSSe + OH, $CSe_2$ + OH, and $H_2Se$ + OH). These reaction rate constants were estimated in Fig. 2, which shows the ratio of the Se compounds' rates to analogous S rates (i.e., an enhancement factor for replacing an S atom with Se) plotted versus the S reaction rate. For S reactions which have a fast reaction rate (e.g., DMS +

Cl, $k = 1.8 \times 10^{-10}$ $cm^3$ $molec^{-1}$ $s^{-1}$), replacing S with Se does not yield a large difference in measured rates (DMSe + Cl, $k = 5.0 \times 10^{-10}$ $cm^3$ $molec^{-1}$ $s^{-1}$). This is because these reactions are already close to the collision-controlled limit, and thus lowering the activation energy by substituting a Se atom for S has little impact on the overall rate. On the other hand, slow reactions like DMS + $O_3$ are sped up by more than four orders of magnitude when Se is substituted for S. We used the log-log relationship in Fig. 2 to predict the reaction rates for OCSe, CSSe, and $H_2Se$ with OH (blue stars). The $CSe_2$ reaction with OH

is calculated from the CSSe reaction, assuming a similar enhancement for the substitution of a second Se atom as between the measured $CSe_2$ + O and CSSe + O reaction rates (Li et al., 2005). The branching ratio for the CSSe + OH reaction products was assumed to be 30% OCSe and 70% OX_Se_IN, the same as the measured CSSe + O branching ratio (Li et al., 2005). We recognize that these estimates are inherently uncertain, and therefore address these uncertainties in our sensitivity analysis (Sect. 3.1.2).

The photolysis of gas-phase Se compounds was included using absorption cross sections of $H_2Se$ (Goodeve and Stein, 1931) and OCSe (Finn and King, 1975). The absorption cross section of $CSe_2$ (King and Srikameswaran, 1969) has been measured, however in too low resolution to be incorporated into the model. Therefore, we assume that $CSe_2$ and CSSe have the same





**Table 2.** Rate constants of Se compound gas-phase reactions at around 298 K and the corresponding rate constant of the analogous S compound. All S reaction rates are from Burkholder et al. (2015), except the DMS + $O_3$ reaction rate which is taken from Wang et al. (2007). No corresponding rate constants for $CSe_2$ reactions are listed, since $CSe_2$ is obtained from doubly substituting Se in $CS_2$.

| Reaction | Se rate constant $(cm^3\,molec^{-1}\,s^{-1})$ | Corresponding S constant $(cm^3\,molec^{-1}\,s^{-1})$ | Reference for Se rate constant |
|---|---|---|---|
| **Measured reactions** | | | |
| $OCSe + O \rightarrow CO + OX\_Se\_IN$ | $2.4 \times 10^{-11}$ | $1.3 \times 10^{-14}$ | Li et al. (2005) |
| $CSe_2 + O \rightarrow OCSe + OX\_Se\_IN$ (32%) $\rightarrow 2\,OX\_Se\_IN$ (68%) | $1.4 \times 10^{-10}$ | - | Li et al. (2005) |
| $CSSe + O \rightarrow OCSe$ (30%) $\rightarrow OX\_Se\_IN$ (70%) | $2.8 \times 10^{-11}$ | $3.6 \times 10^{-12}$ | Li et al. (2005) |
| $DMSe + OH \rightarrow OX\_Se\_OR$ | $6.8 \times 10^{-11}$ | $6.7 \times 10^{-12}$ | Atkinson et al. (1990) |
| $DMSe + NO_3 \rightarrow OX\_Se\_OR$ | $1.4 \times 10^{-11}$ | $1.1 \times 10^{-12}$ | Atkinson et al. (1990) |
| $DMSe + O_3 \rightarrow OX\_Se\_OR$ | $6.8 \times 10^{-17}$ | $2.2 \times 10^{-21}$ | Atkinson et al. (1990) |
| $DMSe + Cl \rightarrow OX\_Se\_OR$ | $5.0 \times 10^{-10}$ | $1.8 \times 10^{-10}$ | Thompson et al. (2002) |
| $H_2Se + O \rightarrow OX\_Se\_IN$ | $2.1 \times 10^{-12}$ | $2.2 \times 10^{-14}$ | Agrawalla and Setser (1987) |
| $H_2Se + Cl \rightarrow OX\_Se\_IN$ | $5.5 \times 10^{-10}$ | $7.4 \times 10^{-11}$ | Agrawalla and Setser (1986) |
| $H_2Se + O_3 \rightarrow OX\_Se\_IN$ | $3.2 \times 10^{-16}$ | $< 2.0 \times 10^{-20}$ | Belyaev et al. (2012) |
| **Estimated reactions** | | | |
| $OCSe + OH \rightarrow OX\_Se\_IN$ | $5.8 \times 10^{-13}$ | $2.0 \times 10^{-15}$ | Estimated, see text |
| $CSe_2 + OH \rightarrow OCSe + OX\_Se\_IN$ | $1.5 \times 10^{-10}$ | - | Estimated, see text |
| $CSSe + OH \rightarrow OX\_Se\_IN$ (70%) $\rightarrow OCSe$ (30%) | $3.0 \times 10^{-11}$ | $1.2 \times 10^{-12}$ | Estimated, see text |
| $H_2Se + OH \rightarrow OX\_Se\_IN$ | $7.2 \times 10^{-11}$ | $4.7 \times 10^{-12}$ | Estimated, see text |

cross section as $CS_2$ (Burkholder et al., 2015). Given the lack of available information, quantum yields for all Se photolysis reactions were assumed to be 1.

### 2.2.4 Condensation of Se on preexisting aerosol particles

As nonvolatile species, oxidized inorganic and organic Se would condense on available atmospheric surfaces. In the SOCOL-
5 AER model, the uptake of these oxidized Se species by sulfate aerosols is calculated similarly to the existing scheme of gas-phase $H_2SO_4$ uptake on sulfate particles (Sheng et al., 2015). We track the size distribution of Se in the aerosol phase with 40 tracers, one for each sulfate aerosol size bin. The sulfate aerosol size distribution changes through processes like growth, evaporation, and coagulation. We track how these microphysical processes change the size distribution of condensed Se through mass-conserving schemes. Evaporation of condensed Se only occurs when the smallest sulfate aerosol bin evaporates, releasing



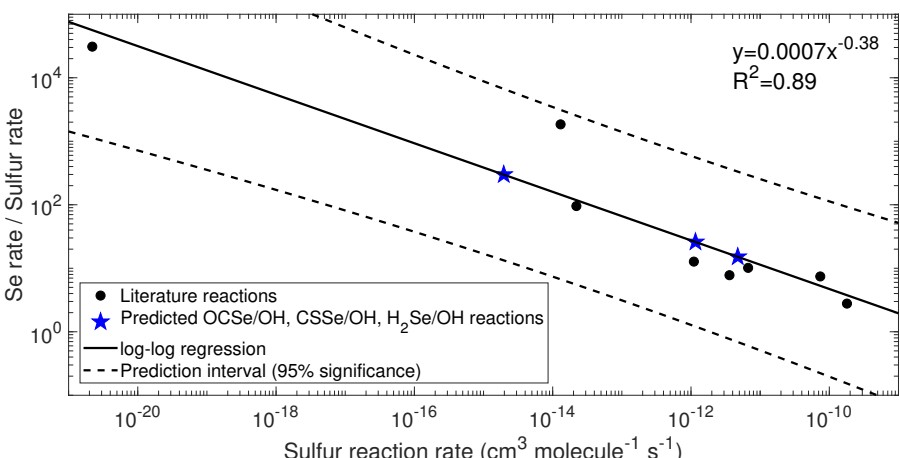

**Figure 2.** Estimation of unknown Se reaction rates from the analogous S reaction rate. A power regression is performed, with statistics shown in the upper right corner of the plot. For the $H_2S + O_3$ reaction only an upper limit estimate was available, and therefore it was not included in the analysis.

the Se stored in that bin as gas-phase inorganic oxidized Se. Sedimentation and deposition of the host sulfate particles are concurrent with sedimentation and deposition of the condensed Se tracers. Gas-phase oxidized Se tracers are also removed by dry and wet deposition, with the assumption that they have the same Henry's law constant as gas-phase $H_2SO_4$.

One limitation of using SOCOL-AER for the Se cycle is that the model only includes online sulfate aerosols. This means

that the transport of Se on other aerosols, including dust, sea salt, and organic aerosols, would be neglected. This may not be a poor assumption, since Se and S are often co-emitted and have been found to be highly correlated in atmospheric aerosol measurements (e.g., Eldred, 1997; Weller et al., 2008). Nevertheless, we included a "dummy" aerosol tracer to test the effect of missing aerosol species in SOCOL-AER. The dummy aerosol tracer represents monodisperse particles that are emitted in a latitudinal band in the model and undergo Se uptake, sedimentation, and wet and dry deposition. This dummy aerosol tracer is

clearly a simplification of true atmospheric processes, as in reality other aerosols are distributed in size and can coagulate with sulfate aerosol particles. However, by varying the radius, location, and emission magnitude of these particles (Sect. 3.1.7), we can determine whether missing aerosols affect atmospheric Se cycling.

## 3  Statistical methods

To conduct the sensitivity analysis of our Se model, we first need to select the input parameters that would be included in

the sensitivity analysis. The probability distributions of these input parameters' uncertainties were determined by reviewing literature sources and using our best judgement. Variance-based sensitivity analysis methods usually require $10^4$ to $10^6$ model runs, which would be prohibitively expensive for the full SOCOL-AER model. We therefore replace the SOCOL-AER model





with a surrogate PCE model. The SOCOL-AER model is run at carefully selected points within the parameter space, creating a set of "training" runs. The training runs are used to produce a surrogate PCE model, which approximates the outputs of the full SOCOL-AER model throughout the input parameter space. Sensitivity indices can then be derived from the surrogate model. All statistical methods presented in this section are available in UQLAB, an open-source MATLAB-based framework

for uncertainty quantification (Marelli and Sudret, 2014).

### 3.1 Uncertainty ranges of input parameters

We restricted the scope of our sensitivity analysis to parameters that have been implemented in the model as part of the Se cycle. We neglect other model uncertainties, including those related to the wet deposition parametrization or the emissions of sulfate aerosol precursors. The focus of our sensitivity analysis is to prioritize which Se-related parameters should be further

constrained. Since we do not vary all other model parameters, the uncertainties of output quantities may be underestimated. However, given the large dimension of our parameter space with 34 Se-related input parameters, including additional non-Se-related parameters would be challenging. In the following section, we will discuss the uncertainty distributions for each of the 34 input parameters included in our study. Due to the lack of detailed information available in literature about the parameter distributions, we chose loguniform or uniform distributions for all but one of the parameters. This follows the conservative

approach recommended by the Maximum Entropy Principle, as the uniform and loguniform distributions maximize entropy while fulfilling the data constraints (Kapur, 1989). The uncertainty distributions of all input parameters are listed in Table 3.

### 3.1.1 Measured rate constants ($k_1$–$k_{12}$)

The Se reactions studied in the literature have each only been measured by one laboratory group (Table 2). Since only one measurement technique has been applied, the reported measurement uncertainties may underestimate the true uncertainties of

these rate constants. To approximate an uncertainty distribution for these rate constants, we reviewed S compound reactions that have been studied by multiple research groups. The reaction that had the largest spread in reported rate constants at ~298 K was OCS + OH, which has been measured in six studies (Atkinson et al., 1978; Kurylo, 1978; Cox and Sheppard, 1980; Leu and Smith, 1981; Cheng and Lee, 1986; Wahner and Ravishankara, 1987; Burkholder et al., 2015). The measured reaction rate constant varied over multiple orders of magnitude; therefore, we calculated its variability on a logarithmic scale. The coefficient

of variation (standard deviation divided by mean) of this reaction rate in logarithmic space was around 6%. We assumed that this maximum S coefficient of variation would apply to the measured Se reaction rates. The bounds were calculated as 88% and 112% (±2 standard deviations) of the available measured rate constant in logarithmic space, i.e.:

$$\text{Bounds} = k^{1 \pm 0.12} \tag{1}$$

where $k$ is the measured rate constant and "Bounds" are its upper and lower bounds, all expressed in $\text{cm}^3\,\text{molec}^{-1}\,\text{s}^{-1}$. The

maximum upper bound was set to $1.0 \times 10^{-9}\ \text{cm}^3\,\text{molec}^{-1}\,\text{s}^{-1}$, since at this order of magnitude the Se reaction rates are collision-limited (Seinfeld and Pandis, 2016).



**Table 3.** Probability distributions of the model input parameters selected for the sensitivity analysis.

| Input parameter | Description | Lower bound | Upper bound | Distribution |
|---|---|---|---|---|
| **Measured reaction rate coefficients ($\mathrm{cm^3\,molec^{-1}\,s^{-1}}$)** | | | | |
| $k_1$ | $OCSe + O \rightarrow CO + OX\_Se\_IN$ | $1.3 \times 10^{-12}$ | $4.5 \times 10^{-10}$ | loguniform |
| $k_2$ | $CSe_2 + O \rightarrow OCSe + OX\_Se\_IN$ | $2.6 \times 10^{-12}$ | $7.9 \times 10^{-10}$ | loguniform |
| $k_3$ | $CSe_2 + O \rightarrow 2\,OX\_Se\_IN$ | $6.0 \times 10^{-12}$ | $1.0 \times 10^{-9}$ | loguniform |
| $k_4$ | $CSSe + O \rightarrow OCSe$ | $4.0 \times 10^{-13}$ | $1.8 \times 10^{-10}$ | loguniform |
| $k_5$ | $CSSe + O \rightarrow OX\_Se\_IN$ | $1.0 \times 10^{-12}$ | $3.9 \times 10^{-10}$ | loguniform |
| $k_6$ | $DMSe + OH \rightarrow OX\_Se\_OR$ | $4.1 \times 10^{-12}$ | $1.0 \times 10^{-9}$ | loguniform |
| $k_7$ | $DMSe + NO_3 \rightarrow OX\_Se\_OR$ | $7.0 \times 10^{-13}$ | $2.8 \times 10^{-10}$ | loguniform |
| $k_8$ | $DMSe + O_3 \rightarrow OX\_Se\_OR$ | $7.8 \times 10^{-19}$ | $5.9 \times 10^{-15}$ | loguniform |
| $k_9$ | $DMSe + Cl \rightarrow OX\_Se\_OR$ | $3.8 \times 10^{-11}$ | $1.0 \times 10^{-9}$ | loguniform |
| $k_{10}$ | $H_2Se + O \rightarrow OX\_Se\_IN$ | $8.3 \times 10^{-14}$ | $5.3 \times 10^{-11}$ | loguniform |
| $k_{11}$ | $H_2Se + Cl \rightarrow OX\_Se\_IN$ | $4.6 \times 10^{-11}$ | $1.0 \times 10^{-9}$ | loguniform |
| $k_{12}$ | $H_2Se + O_3 \rightarrow OX\_Se\_IN$ | $4.4 \times 10^{-18}$ | $2.3 \times 10^{-14}$ | loguniform |
| **Estimated reaction rate coefficients ($\mathrm{cm^3\,molec^{-1}\,s^{-1}}$)** | | | | |
| $k_{13}$ | $OCSe + OH \rightarrow OX\_Se\_IN$ | $2.7 \times 10^{-14}$ | $2.4 \times 10^{-11}$ | loguniform |
| $k_{14}$ | $CSe_2 + OH \rightarrow OCSe + OX\_Se\_IN$ | $7.8 \times 10^{-12}$ | $1.0 \times 10^{-9}$ | loguniform |
| $k_{15}$ | $CSSe + OH \rightarrow OX\_Se\_IN$ | $1.1 \times 10^{-12}$ | $9.3 \times 10^{-10}$ | loguniform |
| $k_{16}$ | $CSSe + OH \rightarrow OCSe$ | $4.7 \times 10^{-13}$ | $4.0 \times 10^{-10}$ | loguniform |
| $k_{17}$ | $H_2Se + OH \rightarrow OX\_Se\_IN$ | $3.7 \times 10^{-12}$ | $1.0 \times 10^{-9}$ | loguniform |
| **Scaling factors for photolysis rates** | | | | |
| $fk_{18}$ | $OCSe + h\nu \rightarrow OX\_Se\_IN$ | 0 | 2 | uniform |
| $fk_{19}$ | $CSe_2 + h\nu \rightarrow 2\,OX\_Se\_IN$ | 0 | 2 | uniform |
| $fk_{20}$ | $CSSe + h\nu \rightarrow OX\_Se\_IN$ | 0 | 2 | uniform |
| $fk_{21}$ | $H_2Se + h\nu \rightarrow OX\_Se\_IN$ | 0 | 2 | uniform |
| **Global emission by source category ($\mathrm{Gg\,Se\,yr^{-1}}$)** | | | | |
| $emiss_{mar}$ | Marine biogenic Se emissions | 0.4 | 35 | loguniform |
| $emiss_{terr}$ | Terrestrial biogenic Se emissions | 0.15 | 5.25 | uniform |
| $emiss_{ant}$ | Anthropogenic Se emissions | 3 | 9.6 | uniform |
| $emiss_{vol}$ | Volcanic Se emissions | 0.076 | 49.1 | loguniform |
| **Speciation of emissions (%)** | | | | |
| $\%OCSe_{ant}$ | OCSe fraction in anthropogenic emissions | 0 | 6 | uniform |
| $\%CSe_{2ant}$ | $CSe_2$ fraction in anthropogenic emissions | 0 | 6 | uniform |
| $\%CSSe_{ant}$ | CSSe fraction in anthropogenic emissions | 0 | 6 | uniform |
| $\%H_2Se_{ant}$ | $H_2Se$ fraction in anthropogenic emissions | 0 | 6 | uniform |
| $\%H_2Se_{vol}$ | $H_2Se$ fraction in volcanic emissions | 0 | 13 | uniform |
| **Accommodation coefficient** | | | | |
| $acc_{coeff}$ | Oxidized Se uptake on aerosols | 0.02 | 1 | uniform |
| **Dummy aerosol parameters** | | | | |
| $r_{aer}$ | Radius of missing aerosol ($\mu$m) | 0.01 | 3 | loguniform |
| $emiss_{aer}$ | Emission magnitude of missing aerosol ($\mathrm{kg\,yr^{-1}}$) | - | - | lognormal (see text) |
| $lat_{aer}$ | Latitudinal band of aerosol emission | 90° to 80° S | 80° to 90° N | uniform |





### 3.1.2 Estimated rate constants ($k_{13}$–$k_{17}$)

Five Se reaction rate constants have not been measured previously in the laboratory and were estimated based on their relationship with analogous S rate constants (Fig. 2). We calculated the uncertainty bounds of estimated rate constants using the 95% error interval of prediction with a linear regression (Wackerly et al., 2014):

$$\text{Bounds} = 10 \char`\^ \left[ x + \hat{Y} \pm t_{0.025} \sqrt{ \frac{\text{SSE}}{n-2} \left( 1 + \frac{1}{n} + \frac{(x-\bar{x})^2}{S_{xx}} \right) } \right] \tag{2}$$

where $x$ is the logarithm (to the base 10) of the corresponding S rate constant, $\hat{Y}$ is the logarithm of the predicted ratio of the Se rate to the S rate, $n$ is the number of data points in the regression, $t_{0.025}$ is the 2.5[th] percentile value of the Student's $t$ distribution for $n-2$ degrees of freedom, $\bar{x}$ is the mean of the logarithm of S rate constants in Fig. 2, SSE is the sum of squares of the residuals, and $S_{xx}$ is the variance of the S rate constants in Fig. 2. All rate constants are in units of $\text{cm}^3 \, \text{molec}^{-1} \, \text{s}^{-1}$.

### 3.1.3 Photolysis rate scaling factors

Uncertainties in our calculated Se photolysis rates arise from uncertainties in the measured OCSe and $H_2$Se cross sections, the assumption that $CSe_2$ and CSSe have the same cross section as $CS_2$, the quantum yields of photolysis reactions, and the look-up table approach that SOCOL-AER applies to calculate photolysis rates. Given the lack of specific information about these uncertainties, we apply a scaling factor on the calculated photolysis rates ranging from 0 to 2 (Table 3).

### 3.1.4 Global emissions from source categories

For the sensitivity analysis, we do not alter the spatial distribution of Se emissions from anthropogenic, marine biogenic, terrestrial biogenic, and volcanic sources (Fig. 3). The parameters that we varied are the scaling factors for each map, i.e., the global total emissions from each source. We reviewed atmospheric Se budget estimates to determine the range in total emissions for different sources. The estimates for global DMSe emissions ranged from the lower limit value of $0.4 \, \text{Gg Se yr}^{-1}$ in Nriagu (1989) to $35 \, \text{Gg Se yr}^{-1}$ in Amouroux et al. (2001). DMSe emissions are calculated online in the model from a marine DMSe concentration map. Based on the results of a previous simulation, we normalize the marine DMSe concentration map in Fig. 3a so that it leads to $1 \, \text{Gg Se yr}^{-1}$ emissions globally. All other maps are also normalized to $1 \, \text{Gg Se yr}^{-1}$ emissions, so that we can directly apply the total global emissions as a scaling factor. The widest uncertainty range of terrestrial emissions was given by Nriagu (1989), from $0.15$–$5.25 \, \text{Gg Se yr}^{-1}$. Total global anthropogenic Se emissions in 1983 were estimated between 3.0 and $9.6 \, \text{Gg Se yr}^{-1}$ (Nriagu and Pacyna, 1988). We applied the same uncertainty range to the total anthropogenic emissions in the year 2000, because it is unclear how global Se emissions have changed during this period. To estimate global Se emissions from degassing volcanoes, we reviewed measurements of Se to S ratios in volcanic emissions, extending the studies reviewed in Floor and Román-Ross (2012) (Table S1). There is a high degree of variability in the emitted Se to S ratios between different volcanoes, and even temporally and spatially for the same volcano (Floor and Román-Ross, 2012). The Se to S ratio in volcanic emissions ranges from $6 \times 10^{-6}$ for White Island, New Zealand (Wardell et al., 2008) to $3.9 \times 10^{-3}$ for Merapi Volcano, Indonesia (Symonds et al., 1987). By multiplying this range in ratios with the global mean total degassing



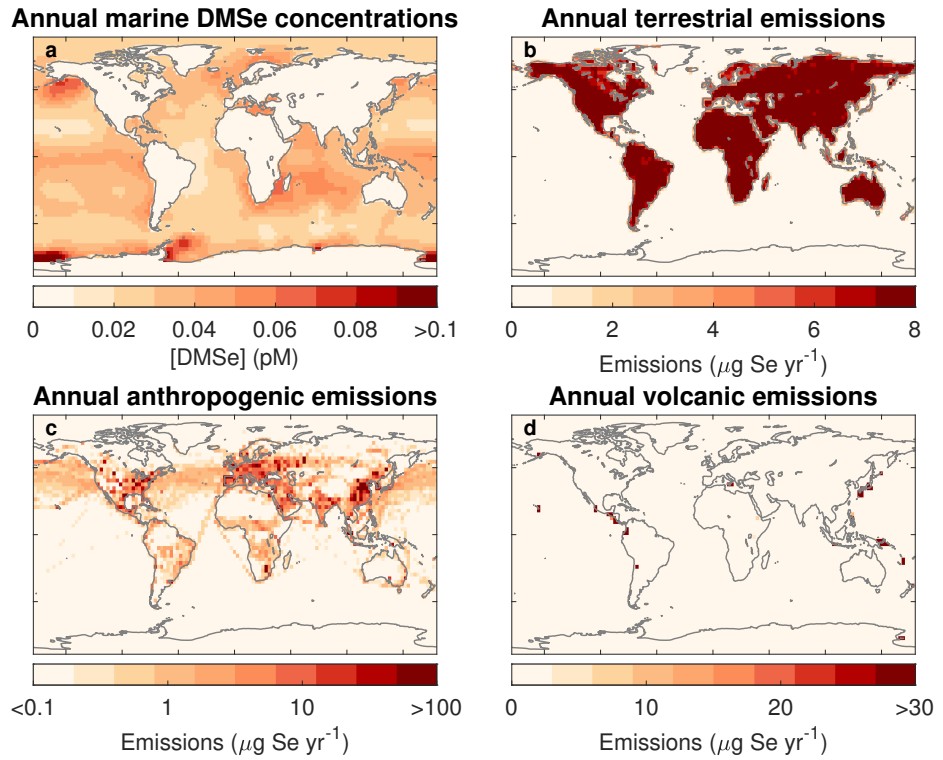

**Figure 3.** Spatial distribution of Se sources. Each source map is normalized so that $1\,\mathrm{Gg\,Se\,yr}^{-1}$ would be emitted globally. This is scaled during the sensitivity analysis.

$SO_2$ emissions from Andres and Kasgnoc (1998) and Dentener et al. (2006), $12.6\,\mathrm{Tg\,S\,yr}^{-1}$, we calculate an uncertainty range for total volcanic Se emissions: $0.076$–$49.1\,\mathrm{Gg\,Se\,yr}^{-1}$. Loguniform distributions were used for the source types whose total emission ranges vary by more than 2 orders of magnitude (volcanic and marine biogenic), whereas uniform distributions were used for the terrestrial and anthropogenic Se emissions.

5   **3.1.5   Speciation of Se emission sources**

The available speciation information for Se emissions is largely qualitative; possible emission species have been identified but not quantified (Sect. 2.2.2). To estimate the uncertainty range for Se emission speciation, we use estimates of speciation from an atmospheric S budget (Watts, 2000). The second most important anthropogenic S species after $SO_2$ is $H_2S$. Watts (2000) estimates the anthropogenic emissions of $H_2S$ are $3.1 \pm 0.3\,\mathrm{Tg\,S\,yr}^{-1}$, compared to an anthropogenic $SO_2$ emission

10   total of $53.2\,\mathrm{Tg\,S\,yr}^{-1}$ (Lamarque et al., 2010; Smith et al., 2011) in the year 2000. $H_2S$ therefore contributes at most 6% of total S emissions. Since $OCSe$, $CSe_2$, $CSSe$, and $H_2Se$ are in general less stable than the analogous S species (Table 2), we consider 6% to be a maximum value for the mass fraction of anthropogenic Se emissions that come from each of these





species. The anthropogenic speciation fractions for OCSe, $CSe_2$, CSSe, and $H_2Se$ are varied between 0 and 6%. The rest of the anthropogenic emissions, 76–100%, are attributed to OX_Se_IN, representing species such as Se and $SeO_2$.

Regarding the speciation of volcanic S emissions, Watts (2000) estimates that $0.99 \pm 0.88 \, \mathrm{Tg\,S\,yr^{-1}}$ is in the form of $H_2S$. Comparing this to the estimate for volcanic $SO_2$ emissions, $12.6 \, \mathrm{Tg\,S\,yr^{-1}}$ (Andres and Kasgnoc, 1998; Dentener et al.,

2006), $H_2S$ contributes at most 13% of volcanic S emissions. Therefore, in our sensitivity analysis the percentage of volcanic Se emissions in the form of $H_2Se$ ranges from 0–13%. Conversely, the percentage of OX_Se_IN in volcanic emissions ranges from 87–100%.

### 3.1.6   Accommodation coefficient of oxidized Se

The accommodation coefficient represents the probability that a gas-phase oxidized Se molecule will stick to an aerosol upon

collision. No laboratory studies have investigated the accommodation coefficient of oxidized Se on aerosol surfaces. However, review papers suggest that Se efficiently partitions to the aerosol phase upon oxidation (Mosher and Duce, 1987; Wen and Carignan, 2007), indicating a high accommodation coefficient. Due to the lack of further information, we assume an uncertainty range of 0.02–1 for the accommodation coefficient, as selected by Lee et al. (2011) for $H_2SO_4$. This accommodation coefficient applies to uptake of Se on sulfate and dummy aerosols.

### 15   3.1.7   Dummy aerosol tracers

SOCOL-AER only includes sulfate aerosol, lacking other aerosol types (e.g., dust, sea salt, organic aerosol, etc.) that may also transport Se. To test how these other aerosol types might affect Se transport and deposition, in the sensitivity analysis we vary the emission location, particle radius, and emission magnitude of the dummy aerosol tracer (introduced in Sect. 2.2.4). These input parameters are different from other uncertainties, in that they are intended to investigate a lack in model

completeness rather than uncertainties in measurable quantities. In our experiments the aerosols are emitted in one of 18 latitude bands, ranging from 90°–80° S to 80°–90° N. The latitude parameter can demonstrate whether Se deposition is affected by these missing aerosol types only in certain latitudinal bands. The emission radius affects the transport of Se on these missing particles, since the atmospheric lifetime of these particles depends on radius (Seinfeld and Pandis, 2016). For example, additional coarse particles (radius > 1 μm) could inhibit far-range Se transport, since they sediment and are removed quicker

than the accumulation mode sulfate particles. We vary the emission radius between 0.01 and 3 μm, as this covers the range of atmospherically relevant particle sizes.

To determine a reasonable range for the emission magnitude of additional aerosol particles, we analyzed the particle emission inventories from the AEROCOM I project (Dentener et al., 2006). Aerosol types are segregated into size classes based on their effective radius, and the emission magnitude for 10° latitude bands is calculated for each size class. The mass emission of

particles correlates with the particle size, with generally larger mass emissions for larger particle sizes (Fig. S1). Therefore, the value of emission magnitude in our experiments depends lognormally on the particle radius ($r$), with mean $\mu = 2.54r + 19.0$ and standard deviation $\sigma = 2.88$. We acknowledge that in the real atmosphere particles are emitted globally as size distributions, not monodisperse particles in one latitudinal band. Nevertheless, by including these input parameters as part of the sensitivity





analysis, we can identify whether the lack of tropospheric aerosols other than sulfate impacts the simulated Se deposition maps in SOCOL-AER.

### 3.2   Experimental setup and model outputs

The creation of surrogate models requires a set of training runs with the full SOCOL-AER model. The values of the input
parameters are varied simultaneously between training runs, so that interactions between parameters can also be detected. A Latin hypercube design is used to draw $N$ samples from an $M$-dimensional input parameter space. In Latin hypercube sampling, each parameter range is divided into $N$ equally probable sub-intervals. $N$ values for each parameter are drawn randomly from within each sub-interval. The selected values for all input parameters are then matched randomly with each other, to yield $N$ points that cover the parameter space better than purely random Monte Carlo sampling (McKay et al., 1979).
The general rule of thumb is to select around $10M$ training simulations, to adequately cover the sample space (Loeppky et al., 2009). We ran $N = 400$ training simulations in our sensitivity analysis with 34 input parameters. Producing training simulations from the full SOCOL-AER model is the most computationally expensive step in the uncertainty and sensitivity analysis. With 48 cores each training run takes around 14 hours, corresponding to around $10^9$ core-seconds for 400 training simulations.

The initial conditions file for the training runs was created from a previous 10 year spinup of the model under year 2000
conditions. The atmospheric mixing ratios of Se tracers are initially set to 0 in each simulation. The simulations are 18 months long, with the first 6 months considered to be an equilibrium period for the Se tracers. Therefore, we analyze only the last 12 months of the 18-month simulation. The model is run with T42 horizontal resolution (approximately $2.8° \times 2.8°$ or $300 \text{ km} \times 300 \text{ km}$ in latitude and longitude) with 39 vertical levels up to 0.01 hPa ($\sim$80 km). The interactions between chemical species (i.e., greenhouse gases and aerosols) and radiation are decoupled in our simulations. The decoupling between
chemistry and climate prevents meteorological differences between training runs with different input parameters, eliminating the influence of precipitation variability on deposition. Deposition flux differences in these relatively short simulations can then be more easily attributed to changes in the input parameters.

All relevant Se cycle fluxes and reservoir burdens are outputted by SOCOL-AER as monthly averages. For the sensitivity analysis, the target outputs are annual mean values of the global atmospheric Se lifetime and Se surface deposition fluxes.
The global Se lifetime is calculated as the total atmospheric Se burden divided by total deposition (Seinfeld and Pandis, 2016). Deposition fluxes of Se were calculated by summing up the dry and wet deposition fluxes of aerosol and gas-phase Se species. Since the model is run in T42 horizontal resolution, there are 8192 surface grid boxes representing geographical coordinates on the globe. We therefore have 8192 model outputs for Se deposition and we construct a PCE model and conduct a sensitivity analysis for each one. It can be argued that the computational cost can further be reduced by dimensionality
reduction techniques (e.g., Blatman and Sudret, 2011b, 2013; Ryan et al., 2018), like building the PCE models on a reduced output set coming from a principal component analysis (PCA). We did not consider such an approach in this work because the cost of building the 8192 PCE models is marginal compared to that of evaluating the full SOCOL-AER model. In summary, the 400 SOCOL-AER training runs yield 400 values for atmospheric Se lifetimes and $400 \times 8192$ values for deposition fluxes.





### 3.3 Surrogate modeling with PCE

Sudret (2008) provided a detailed description of using PCE to build surrogate models, which was updated by Le Gratiet et al. (2017). We will summarize the most important features of the approach here. For this study, a certain output of the SOCOL-AER model ($Y$) can be thought of as a finite variance random variable which is a function of the $M = 34$ input parameters ($\mathbf{X} = [X_1, X_2, \ldots, X_{34}]$):

$$Y = \mathcal{M}(\mathbf{X}) \tag{3}$$

The input $\mathbf{X} \in \mathbb{R}^M$ is modeled by a joint probability density function (PDF) $f_{\mathbf{X}}$ whose marginals are assumed independent, i.e., $f_{\mathbf{X}} = \prod_{i=1}^{M} f_{X_i}(x_i)$.

In polynomial chaos decomposition, the output variable $Y$ is decomposed into the following infinite series (Ghanem and Spanos, 2003):

$$Y = \sum_{\boldsymbol{\alpha} = \mathbb{N}^M} y_{\boldsymbol{\alpha}} \psi_{\boldsymbol{\alpha}}(\mathbf{X}) \tag{4}$$

where $y_{\boldsymbol{\alpha}}$ are coefficients to be determined and $\boldsymbol{\alpha}$ is a multi-index set that defines the degree of the multivariate orthonormal polynomial $\psi_{\boldsymbol{\alpha}}(\mathbf{X}) = \prod_{i=1}^{M} \psi_{\alpha_i}(X_i)$. The latter belongs to a family of polynomials that are orthogonal with respect to the marginal PDF $f_{X_i}$. For example, univariate uniform probability distributions correspond to the family of Legendre polynomials and Gaussian probability distributions correspond to the family of Hermite polynomials (Xiu and Karniadakis, 2002). Multivariate orthogonal polynomials can be constructed through multiplying the relevant univariate polynomials. The *order* of a polynomial term is defined as the total number of variables included in the polynomial term. The *degree* of a polynomial term represents the sum of the exponents of all variables appearing in the term. The number of coefficients to estimate grows exponentially with both the dimension and the degree. To allow calculation of the coefficients, the terms in Eq. 4 are truncated by restricting the maximum degree of the polynomials. Advanced truncation schemes can also be used to reduce the number of terms and thus the computational budget. Here we consider hyperbolic truncation which removes high order interaction terms from the PCE, while retaining high degree polynomials of a single variable (Blatman and Sudret, 2011a). Hyperbolic truncation is based on selecting only the terms that satisfy:

$$\mathcal{A} = \left\{ \boldsymbol{\alpha} : \left( \sum_{i=1}^{M} \alpha_i^q \right)^{1/q} \leq p \right\} \tag{5}$$

where $q$ is a value selected between 0 and 1, and $p$ is a value selected as the maximum degree of the PCE. Setting $q = 1$ corresponds to the standard truncation scheme where all terms with degree below $p$ are selected. The lower the value of $q$, the more higher order interaction terms are removed from the PCE. In our case, we selected a $q$ value of 0.75.

PCE coefficients are generally calculated by least-square regression (Berveiller et al., 2006) using an *experimental design* which consists of uniformly sampled realizations of the input variables $\mathcal{X} = \{\mathbf{x}^{(1)} \ldots \mathbf{x}^{(N)}\}$ and corresponding model evalua-





tions $\{\mathcal{M}(\mathbf{x}^{(1)})\dots\mathcal{M}(\mathbf{x}^{(N)})\}$, i.e.:

$$y_{\boldsymbol{\alpha}} = \arg\min_{y_{\boldsymbol{\alpha}}\in\mathbb{R}^{\mathrm{card}\mathcal{A}}} \frac{1}{N}\sum_{i=1}^{N}\left(\mathcal{M}(\mathbf{x}^{(i)}) - \sum_{\boldsymbol{\alpha}\in\mathcal{A}} y_{\boldsymbol{\alpha}}\Psi_{\boldsymbol{\alpha}}(\mathbf{x}^{(i)})\right)^2 \tag{6}$$

When the dimension is large, regression techniques that allow for sparsity, i.e., by forcing some coefficients to be zero, are favored. In this work, we consider least-angle regression as proposed in Blatman and Sudret (2011a) and follow the implementation in the UQLAB PCE module (Marelli et al., 2019).

The accuracy of the PCE in representing the full SOCOL-AER model is evaluated with a cross-validation metric named the leave-one-out (LOO) error, $\epsilon_{\mathrm{LOO}}$ (Blatman and Sudret, 2010a). The cross-validation approach removes the need for an independent validation dataset, saving computational expense. The leave-one-out procedure consists of creating a PCE using all but one of the training simulations, $\mathcal{M}^{\mathrm{PC}\backslash i}$. This PCE is then used to predict the output value of the model at the removed training simulation point, $\mathbf{x}^{(i)}$. The process is repeated for all $N$ training points so that a residual sum of squares can be calculated. This residual is then divided by the output variance in the training dataset, yielding the LOO error:

$$\epsilon_{\mathrm{LOO}} = \frac{\sum_{i=1}^{N}(\mathcal{M}(\mathbf{x}^{(i)}) - \mathcal{M}^{\mathrm{PC}\backslash i}(\mathbf{x}^{(i)}))^2}{\sum_{i=1}^{N}(\mathcal{M}(\mathbf{x}^{(i)}) - \hat{\mu}_Y)^2} \tag{7}$$

where $\hat{\mu}_Y$ is the sample mean of the model output in the training dataset. In practice, the LOO error is estimated using a *single* PCE model considering the entire experimental design (Marelli and Sudret, 2014; Blatman and Sudret, 2010b), to avoid the procedure of creating $N$ PCE models. Equation 7 is evaluated as:

$$\epsilon_{\mathrm{LOO}} = \sum_{i=1}^{N}\left(\frac{\mathcal{M}(\mathbf{x}^{(i)}) - \mathcal{M}^{\mathrm{PC}}(\mathbf{x}^{(i)})}{1 - h_i}\right)^2 \Big/ \sum_{i=1}^{N}\left(\mathcal{M}(\mathbf{x}^{(i)}) - \hat{\mu}_Y\right)^2 \tag{8}$$

where $h_i$ is the $i^{\mathrm{th}}$ component of the vector defined by:

$$\mathbf{h} = \mathrm{diag}\left(\mathbf{A}(\mathbf{A}^T\mathbf{A})^{-1}\mathbf{A}^T\right) \tag{9}$$

and $\mathbf{A}$ is the experimental matrix whose components read:

$$A_{ij} = \Psi_j\left(\mathbf{x}^{(i)}\right) \qquad i = 1,\dots,n; \quad j = 0,\dots,P-1 \tag{10}$$

where $P \equiv \mathrm{card}\mathcal{A}$ is the number of terms in the PCE. In our study, we use a degree-adaptive scheme to construct the PCE models sequentially from degree 1 to maximum degree 13. If the LOO error does not decrease over two steps in degree, the algorithm is stopped and the maximum degree is selected as the one with the lowest LOO error. This method reduces the risk of overfitting the training set. The maximum degree of 13 was selected because the PCE becomes too computationally expensive to calculate above this degree. In any case, almost all PCEs calculated in this study are below degree 10.

To improve the accuracy of the approximation, we applied post-processing steps to the construction of PCE models (Table 4). The global atmospheric Se lifetime is a function of the atmospheric Se burden and the total Se deposition. We found improved overall accuracy when separate PCE models were created for the atmospheric burden and total deposition, rather





**Table 4.** Summary of methods to construct surrogate models and calculate Sobol' sensitivity indices of the SOCOL-AER output parameters.

| Output parameter | Surrogate modeling method | Sensitivity analysis method |
|---|---|---|
| Global atmospheric Se lifetime | Constructed a PCE model of atmospheric Se burden and a PCE model of Se deposition flux. Divided the burden PCE by the deposition PCE. | Monte Carlo estimation of Sobol' sensitivity indices |
| Deposition flux of Se (7319 grid boxes) | Constructed PCE models of deposition fluxes in each grid box (LOO error was less than 0.05). | Sobol' sensitivity indices are extracted directly from deposition PCE models |
| Deposition flux of Se (873 grid boxes with LOO > 0.05) | Constructed PCE models of log-transformed deposition fluxes in each grid box. Exponential of the PCE models yields surrogate models for deposition fluxes. | Monte Carlo estimation of Sobol' sensitivity indices |

than first calculating the lifetime for each simulation and constructing a PCE model of the lifetime. To calculate a surrogate model of the Se lifetime, we divided the PCE model of the atmospheric Se burden by the PCE model of total deposition. When calculating the PCE models of Se deposition fluxes, we first used the deposition fluxes directly from the training simulations. However, 873 of the 8192 grid boxes showed LOO errors higher than 0.05, which was selected as the acceptable threshold for this study. Improved LOO errors were achieved when the simulated deposition fluxes were log-transformed before constructing the PCE model. However, after log-transformation the sensitivity indices for deposition cannot be extracted analytically from the PCE (Sect. 3.4), increasing computational expense. Therefore, we only log-transformed the data from these 873 grid boxes before creating their PCE models. Surrogate models for deposition fluxes in these 873 grid boxes are calculated by taking the exponential of the PCE model of log-transformed data.

## 3.4   Sensitivity analysis

A Sobol' sensitivity index represents the fraction of model variance caused by the parametric uncertainty of a certain input variable or the interaction between multiple variables (Sobol, 1993; Marelli et al., 2019). It is a global measure, meaning that the sensitivity index considers the entire parametric uncertainty space. Let us consider a model $\mathcal{M}$ whose inputs, defined over a domain $\mathcal{D}_{\mathbf{x}}$, are assumed independent. It can be shown that it admits the following so-called Sobol' decomposition:

$$\mathcal{M}(x) = \mathcal{M}_0 + \sum_{i=0}^{M} \mathcal{M}_i(x_i) + \sum_{1 \leq i \leq j \leq M}^{M} \mathcal{M}_{ij}(x_i, x_j) + ... + \mathcal{M}_{1,2,...,M}(x_1,...,x_M) \tag{11}$$

where $\mathcal{M}_0$ is a constant and the other summands are defined such that their integrals with respect to any of their arguments is 0, i.e.:

$$\int_{\mathcal{D}_{\mathbf{x}}} M_{i_1,...,i_s}(x_{i_1},...,x_{i_s}) f_{\mathbf{x}_{i_k}} \mathrm{d}x_{i_k} = 0, \qquad 1 \leq k \leq s \tag{12}$$





Following this decomposition, it can be shown that the variance of the random variable $Y = \mathcal{M}(\mathbf{X})$ can be cast as (Saltelli et al., 2008):

$$D = \text{Var}[\mathcal{M}(\mathbf{X})] = \sum_{i=1}^{M} D_i + \sum_{1 \leq i \leq j \leq M}^{M} D_{ij} + ... + D_{12...M} \tag{13}$$

where a partial variance with respect to several variables $X_{i_1}, \ldots, X_{i_s}$ can be calculated as:

$$D_{i_1,...,i_s} = \int_{\mathcal{D}_{\mathbf{x}}} \mathcal{M}^2_{i_1,...,is}(x_{i_1},...,x_{i_s}) f_{X_{i_1}}(x_{i_1})...f_{X_{i_s}}(x_{i_s}) \mathrm{d}x_{i_1}...\mathrm{d}x_{i_s} \tag{14}$$

The Sobol' indices ($S_{i_1,...,i_s}$) for a subset of input parameters are eventually obtained by normalizing the corresponding variance, i.e.:

$$S_{i_1,...,i_s} = \frac{D_{i_1,...,i_s}}{D} \tag{15}$$

The number of variables involved in a Sobol' index determines its order. A first order Sobol' index ($S_i = \frac{D_i}{D}$) apportions the variance in the model output to the effect of a single variable, $X_i$. Second order indices ($S_{ij} = \frac{D_{ij}}{D}$) represent the impact of the interaction of two variables (e.g., $X_i$ and $X_j$, $i \neq j$) on the model output variance, not already accounted for by $S_i$ and $S_j$. Higher order indices can be calculated as well. The summation of all individual Sobol' indices yields a value of 1, i.e., accounting for all of the variance in the output. However, the number of higher order indices to calculate can become very large:

$$\text{Number of n}^{\text{th}}\text{ order indices} = \binom{n}{M} = \frac{n!}{M!(n-M)!} \tag{16}$$

It therefore becomes computationally demanding in our case with $M = 34$ input parameters to calculate sensitivity indices with order higher than 2. Furthermore, by the *sparsity-of-effect principle* (Goupy and Creighton, 2006), it is expected that the model is primarily driven by first order effects and low order interactions. We therefore only calculate individually the first and second order Sobol' indices, which is common practice in global sensitivity analysis.

The total Sobol' index ($S_i^T$) summarizes all sensitivity indices which include the effect of a given input variable:

$$S_i^T = S_i + \sum_{j \neq i} S_{ij} + \sum_{j \neq i} \sum_{\substack{k \neq i \\ k \neq j}} S_{ijk} + ... \tag{17}$$

In practice, it is possible to calculate the total Sobol' index without individually calculating all of the higher order indices (Marelli et al., 2019). The total Sobol' indices can be used to rank the influence of input variables on the variability in the model output. It should be noted that the sum of all total Sobol' indices will be greater than 1 if the model is non-additive, since interaction terms would be counted multiple times.

Other studies have emphasized the computational expense of conducting a sensitivity analysis for all grid boxes in a chemistry–climate model (Ryan et al., 2018). By using PCE as the surrogate model, we can reduce the computational expense since it is possible to compute the Sobol' sensitivity indices *analytically*. As shown by Sudret (2008), Sobol' sensitivity





indices are a function of the calculated coefficients in the PCE model (Eq. 4). However, this is no longer possible when post-processing steps are applied to the original PCE models to create surrogate models. For example: 1) the surrogate model for the atmospheric lifetime is calculated from two PCE models, one for total Se burden and one for total Se deposition; 2) Se deposition in 873 grid boxes is calculated using the exponential of a PCE model (see Table 4). In these cases, it is still possible

to calculate the Sobol' indices through Monte Carlo estimation (Marelli et al., 2019). To accomplish this, the surrogate models are sampled around $10^6$ times, which remains tractable.

To summarize categories of input variables, we aggregate the Sobol' indices of several input parameters to yield a total Sobol' index for that category. For example, we summarize the total dummy aerosol effect by summing the total Sobol' indices of the dummy aerosol radius, emission magnitude, and latitude. It would anyways be difficult to separate the effects of the dummy

aerosol input parameters since they are correlated inputs in the experimental design. The second order indices involving two dummy aerosol input parameters may be double-counted with this method. However, since these indices are small ($< 0.05$) we do not expect a large error in the aggregated total Sobol' index (Sect. 5.1).

### 3.5 Resampling of surrogate models

In order to estimate distribution statistics (mean, standard deviation, quantiles) of the output variable, we resample each sur-

rogate model 40 000 times. We also use these 40 000 samples of the parameter space to calculate relationships between input parameters and output variables. To visualize marginal relationships between a certain input parameter and the output, we replace the value of the input parameter in the 40 000 samples by a fixed value and calculate the mean and variance of the surrogate model output. Repeating this step with other evenly spaced values in the input parameter range, we can produce the univariate relationship between the model output and the input parameter.

## 4   Compilation of Se wet deposition flux data

We decided to compare SOCOL-AER results with measurements of Se wet deposition, since wet deposition contributes an estimated 80% of total deposition (Wen and Carignan, 2007). We conducted a systematic literature review to assemble a dataset of measured wet Se deposition fluxes, extending an earlier review from Conde and Sanz Alaejos (1997). We searched in Web of Science (Clarivate Analytics) for combinations of the following criteria: "Selenium", "Se", "rain", "precipitation", "wet

deposition", "trace element". The last search was completed in July 2019, yielding a total of 672 search results. We screened these search results for studies that measured Se concentrations in rainwater for at least one month, neglecting studies that measured bulk (wet and dry) deposition or that extrapolated wet deposition fluxes from aerosol measurements. The compiled dataset, which is available in the Supplement, consists of 29 papers and data from two measurement networks, the European Monitoring and Evaluation Programme (EMEP) and Canadian National Atmospheric Chemistry (NAtChem) database, for a

total of 73 measurement sites (Table 5). From these studies, we extracted the annual mean Se deposition fluxes, if available, or we used rainwater volume-mean weighted Se concentrations combined with the annual precipitation depths to calculate the deposition flux. If the paper did not provide the annual precipitation depth, we calculated the mean annual precipitation depth



**Table 5.** Previous studies measuring Se wet deposition fluxes.

| Reference | Location | Reference | Location |
|---|---|---|---|
| Suzuki et al. (1981) | Tokyo, Japan | Arimoto et al. (1985) | Marshall Islands |
| Cutter and Church (1986) | Delaware and Bermuda | Arimoto et al. (1987) | American Samoa |
| Dasch and Wolff (1989) | Massachusetts, USA | Heaton et al. (1990) | Rhode Island, USA |
| Cutter (1993) | Bermuda | Scudlark et al. (1994) | Maryland, USA (2 sites) |
| Al-Momani et al. (1997) | Antalya, Turkey | Cutter and Cutter (1998) | Mace Head, Ireland |
| Lawson and Mason (2001) | Maryland, USA | De Gregori et al. (2002) | Valparaiso, Chile |
| Scudlark et al. (2005) | Maryland, USA | Sakata et al. (2006) | Japan (10 sites) |
| Shimamura et al. (2007) | Tokyo, Japan | Landing et al. (2010) | Florida, USA |
| Zhou et al. (2012) | Mt. Heng, China | Liu et al. (2012) | Shigatse, Tibet |
| Gratz et al. (2013) | Illinois, USA (4 sites) | Pan and Wang (2015) | China (10 sites) |
| Lynam et al. (2015) | Alberta, Canada | Xing et al. (2017) | Jiaozhou Bay, China |
| Nie et al. (2017) | Mt. Lushan, China | Blazina et al. (2017) | Plynlimmon, UK |
| Savage et al. (2017) | Bangladesh and Sri Lanka | Uchiyama et al. (2019) | Tokyo, Japan |
| Suess et al. (2019) | Pic du Midi, France | Pearson et al. (2019) | Alaska, USA (3 sites) |
| Savage et al. (2019) | Northern Ireland, UK | NAtChem database | Ontario, Canada (9 sites) |
| EMEP network | 9 sites in Germany and Czech Republic | | |

for the time period and location of the study from the historical Multi-Source Weighted-Ensemble Precipitation (MSWEP) dataset (Beck et al., 2019). The calculated annual deposition fluxes for 9 sites are extrapolated from less than 12 months of rainwater Se measurements; the majority of sites (64) were measured for longer than a year. For studies that spanned multiple years, a multi-annual mean deposition flux was calculated to compare with the model. Additional metadata was extracted from the papers, including geographic coordinates, the time period, collection methods, sampling frequency, and analytical methods. Despite the fact that the model is based on year 2000 conditions for emission maps and meteorology, we compare the model with data from all measurement years, since the dataset is relatively small. We created surrogate models of the Se wet deposition flux in the grid boxes where measurements were made (as in Sect. 3.3 for the total deposition fluxes). We can then compare the resampled distribution of Se wet deposition in these model grid boxes with the corresponding observation.

# 5 Results

## 5.1 Atmospheric Se lifetime

From the 400 training runs of SOCOL-AER, we created PCE models of the global and annual mean total Se burden and deposition flux. The LOO error of the global burden is around 0.02 and the LOO error of the deposition flux is on the order of $10^{-6}$. The LOO error is low for total deposition since for a mass conserving model total deposition should equal the sum of the





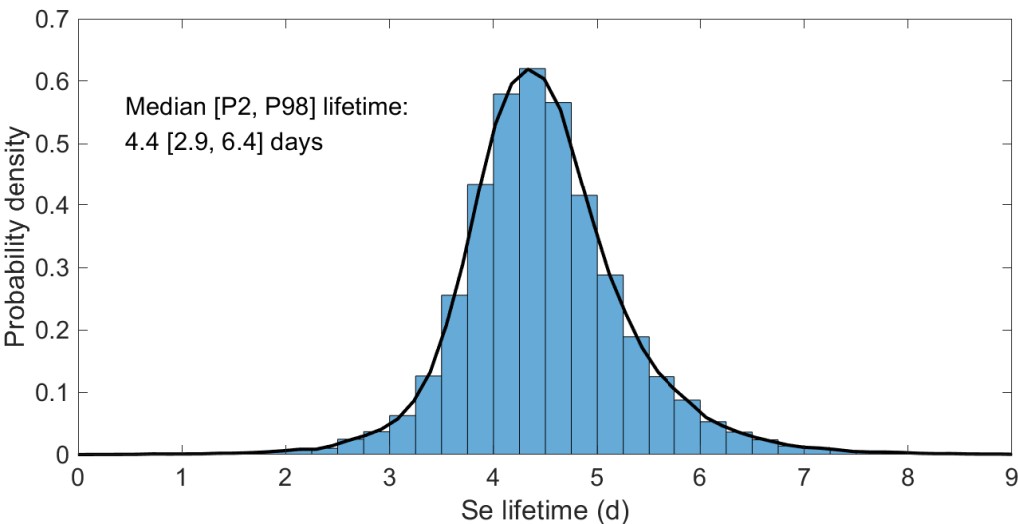

**Figure 4.** Distribution of the atmospheric Se lifetime, resampled from the surrogate model. Summary statistics (median, 2[nd] percentile, and 98[th] percentile value) are listed on the plot.

emission input parameters. Indeed, all 400 training runs showed very good Se mass conservation: total annual Se deposition fluxes were 98–102% of total emission fluxes. We derive a surrogate model for the atmospheric Se lifetime by dividing the Se burden PCE model by the Se deposition PCE model (Sect. 3.3). This surrogate model is resampled to calculate the probability distribution for the atmospheric lifetime, given our assumptions for the uncertainty ranges of all 34 parameters (Sect. 3.5). The

histogram of the atmospheric lifetime (Fig. 4) shows a narrow range for the atmospheric Se lifetime, with median of 4.4 d (days), 2[nd] percentile value of 2.9 d, and 98[th] percentile value of 6.4 d. Despite the large uncertainty ranges for the reaction rate constants of Se, spanning multiple orders of magnitude (Table 3), the uncertainty of the simulated atmospheric lifetime is less than one order of magnitude.

In order to identify the input parameters that drive the variability in the simulated Se lifetime, we apportion the variance into

contributions from the most important parameters (Fig. 5). The most important variable is $k_{13}$, which is the rate constant for the OCSe + OH reaction, followed by the dummy aerosol input parameters. Nonlinearities are also clearly important for the global Se lifetime, since all first order terms only account for 53% of the variance in the Se lifetime, with interaction terms accounting for the other 47%. However, the interaction contribution is made up of many small individual interaction terms. Only two interaction terms account for more than 5% of the total variance in the Se lifetime: the interaction between $k_1$ and $k_{13}$

(5.3% of variance) and the interaction between the dummy aerosol radius and dummy aerosol emissions (5.0% of variance). Through resampling the surrogate model for the Se lifetime, we investigate the relationships between the Se lifetime and input parameters in Figs. 6 and 7.

Several of the most influential input parameters for the Se lifetime are related to OCSe processes (Fig. 6a–c). Given the median estimates for the reaction rate constants in Table 2, OCSe has the longest lifetime of any gas-phase Se species. Therefore,





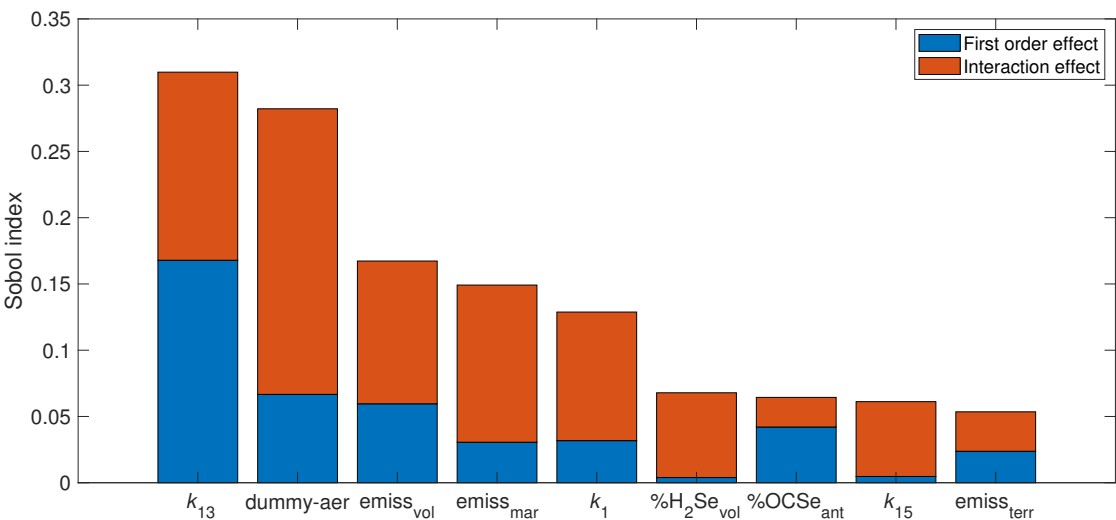

**Figure 5.** Sensitivity indices of the most important parameters ($S_i^T > 0.05$) for the atmospheric Se lifetime. Total Sobol' indices are partitioned into first order and interaction effects. The total Sobol' indices for the dummy aerosol parameters are aggregated, since it is difficult to separate the effects of the correlated input parameters (aerosol radius and emission magnitude).

chemical reaction rates and emissions of OCSe have a strong effect on the overall Se burden and Se lifetime. Slower reaction rates of OCSe with OH ($k_{13}$) and O ($k_1$) lead to longer Se lifetimes. Since OH is more prevalent in the lower atmosphere than O, the dependence of the Se lifetime on $k_{13}$ is stronger than the dependence of the lifetime on $k_1$. The influence of $k_{13}$ on the Se lifetime is mostly saturated above $10^{-12}$ cm$^3$ molec$^{-1}$ s$^{-1}$. Above this threshold in $k_{13}$, the OCSe burden becomes a minor

part of the total Se burden (accounting for less than 2% of total Se), and therefore the Se lifetime is not affected by OCSe processes. The second order interaction between $k_{13}$ and $k_1$ is also shown in Fig. 6b. When the reaction of OCSe with OH is fast ($k_{13} > 10^{-12.5}$ cm$^3$ molec$^{-1}$ s$^{-1}$), the value of $k_1$ is not important for the Se lifetime since almost all OCSe will react with OH. In cases when the OCSe + OH reaction is slow ($k_{13} < 10^{-12.5}$ cm$^3$ molec$^{-1}$ s$^{-1}$), the value of $k_1$ has a stronger effect on the Se lifetime since not all OCSe has reacted with OH. The Se lifetime increases for higher fractions of anthropogenic Se

emitted as OCSe, again showing that higher OCSe burdens prolong the atmospheric Se lifetime. OCSe has been associated with anthropogenic emissions by only one study, which inferred its presence from a similarity in boiling point with a detected Se species (Pavageau et al., 2002). OCSe has never been identified in the ambient atmosphere; on the other hand, SOCOL-AER predicts maximum OCSe concentrations of sub-ppt levels, which would be difficult to measure analytically. Still, processes related to OCSe, a highly speculative atmospheric Se compound, contribute to the model's variability in the Se lifetime.

After OCSe, the most important input parameters for the atmospheric Se lifetime are related to dummy aerosols. It is difficult to interpret the univariate effects of aerosol emissions and radius (Fig. 7a, b), since by design the two inputs are correlated. We hypothesize that the influence of dummy aerosols on the Se lifetime is related to the uptake rate on these dummy aerosols,







**Figure 6.** Relationships between the atmospheric Se lifetime and the most important input parameters from Fig. 5. Using resampling techniques (Sect. 3.5), we calculate the mean and standard deviation Se lifetime over the range of input parameters. Interaction effects are illustrated in the $k_1$ plot ($b$), by grouping the samples into cases where $k_{13}$ is high and low.

which would be proportional to the surface area density. The mass emissions of the aerosols are proportional to volume ($r^3$), which when divided by dummy aerosol radius yields a metric that is proportional to the emitted surface area ($r^2$). The Se lifetime increases monotonically with the surface area of emitted dummy aerosols, although the response is not linear (Fig. 7c). We compare this response with the range of available surface area of sulfate aerosol for 10° latitude bands, shaded in yellow.

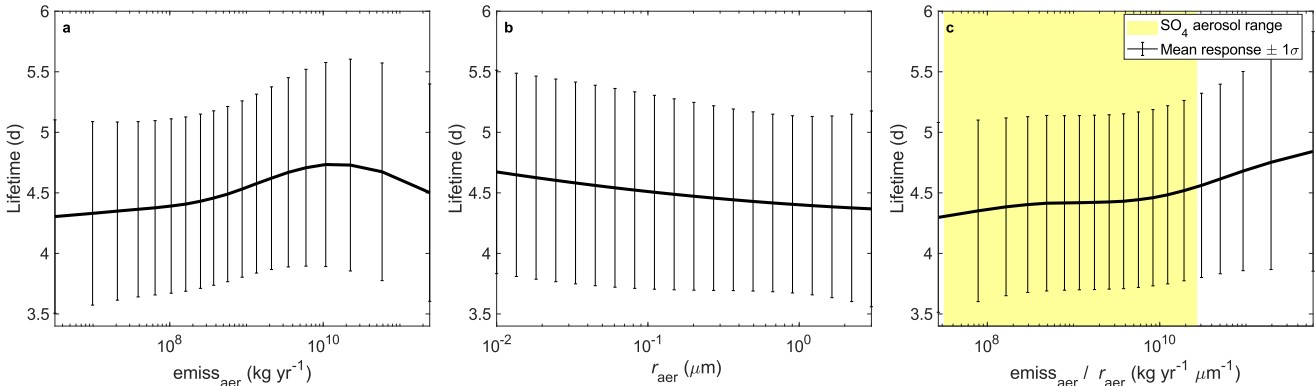

**Figure 7.** Relationship between the atmospheric Se lifetime and the dummy aerosol input parameters (*a-b*). A quantity related to surface area is shown in (*c*), which also includes the comparable range for sulfate aerosol as yellow shading. For sulfate, this is calculated by dividing the deposited mass of sulfate (equal to emitted mass in steady state) in each 10° latitudinal band by the effective sulfate radius in that latitudinal band. Since aerosol mass emissions are a lognormally distributed quantity, the bounds are infinite and thus we include only the 5[th] to 95[th] percentile range in (*a, c*).

For lower dummy aerosol surface areas, the available sulfate aerosol dominates the uptake of gas-phase Se and additional dummy aerosols do not play an important role. The dummy aerosols have a stronger effect on Se lifetime at the upper limit of the sulfate aerosol surface area range. The absence of other aerosols than sulfate leads to a lower Se lifetime in SOCOL-AER. However, the effect of dummy aerosols is not drastic, only increasing the mean Se lifetime from 4.3 to 4.8 d (Fig. 7c).

5      The other inputs have minor impacts on the global Se lifetime. Stronger emissions of volcanic Se lead to shorter overall global Se lifetimes, while emissions of marine and terrestrial biogenic Se lead to longer Se lifetimes (Fig. 6d–f). In SOCOL-AER, biogenic sources emit DMSe, which is not removed by wet and dry deposition, while volcanic sources emit mainly oxidized Se species, which can be removed by wet and dry deposition. Biogenic emissions must first be oxidized before deposition can occur, which can prolong the Se lifetime. The influence of the two other terms with $S_i^T > 0.05$, the reaction rate constant of CSSe + OH ($k_{15}$) and the fraction of volcanic emissions emitted as $H_2Se$, is mainly through interaction terms, and therefore the Se lifetime responds only weakly to univariate variations in these variables (Fig. 6g, h).

## 5.2   Spatial patterns of Se deposition

Surrogate models for Se deposition in all surface grid boxes were calculated according to Sect. 3.3. After log-transforming Se deposition in 873 grid boxes, the LOO error is below 0.1 in all grid boxes, but still above 0.05 in 354 grid boxes, mainly in polar regions (Fig. S2). However, due to limitations in computational time to run more training runs, we proceeded with the sensitivity analyses of all grid boxes. By resampling the surrogate models, we calculate maps of mean Se deposition and its coefficient of variation of over all input uncertainties (Fig. 8). Selenium deposition is highest in areas close to anthropogenic and volcanic emissions (Fig. 3c, d), which are point sources. The deposition pattern is clearly affected by precipitation: dry



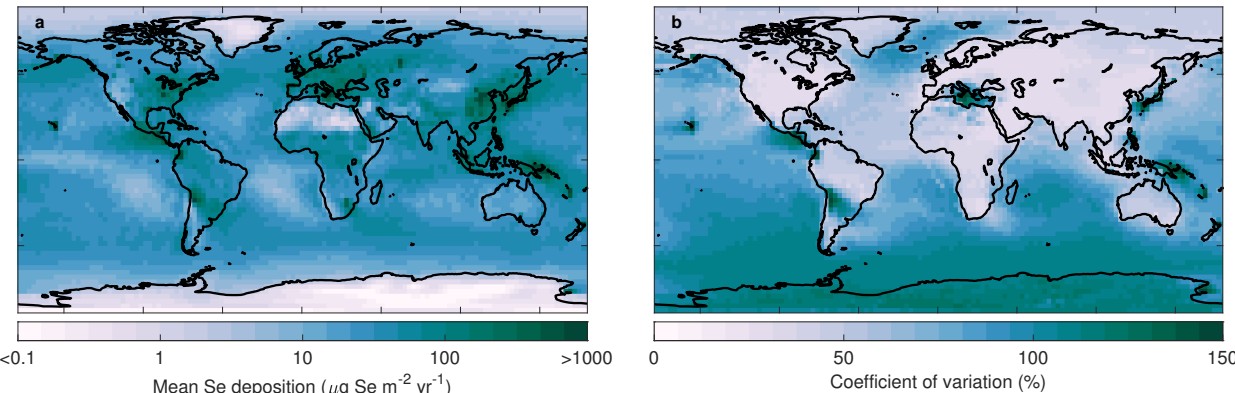

**Figure 8.** Map of the mean Se deposition (*a*) and associated coefficient of variation (*b*), calculated by resampling surrogate models of deposition in each grid box.

areas (e.g., eastern portion of ocean basins, polar regions, and the Sahara Desert) show the lowest Se deposition fluxes globally. Other interesting features of the Se deposition patterns, for example identifying the regions influenced by long-range transport, will be investigated in upcoming studies. In this study, we focus on determining the most important sources of uncertainty to the simulated deposition maps. The relative uncertainty in simulated deposition, illustrated by the coefficient of variation (Fig. 8b),

is highest in areas affected by marine and volcanic emissions because these emission sources have wider uncertainty ranges compared to anthropogenic and terrestrial emissions (Table 3). In the following global sensitivity analysis we can identify the input factors that contribute to the variation in Se deposition in each grid box.

Figures 9 and 10 illustrate the spatial variation in the importance of input parameters. We chose example grid boxes (indicated by blue circles) to illustrate the marginal relationships between Se deposition and the input parameters. Overall, the most

important input parameters are the total emissions from each source. In the example grid boxes shown in Fig. 9a–h, Se deposition increases linearly with increasing the emissions from the different source categories. The linear relationship is logical since deposition balances emission in the steady state. In areas that are more remote from emission regions (Sahara, Antarctic, and Arctic), other factors become more important but are still minor compared to the emission inputs. The aerosol accommodation coefficient affects areas where the precipitation is very low, for example in the Saharan Desert (Fig. 9i). In these dry regions,

total Se deposition is dominated by dry deposition. When the model is run with low accommodation coefficients, less oxidized Se partitions to the particulate phase and more remains in the gas phase. Dry deposition of particles in the 0.1–1 μm diameter size range, within the range of sulfate and dummy aerosols, is slower than gas compounds due to the slower Brownian motion of particles (Seinfeld and Pandis, 2016). Chemical reaction rate constants, specifically the reaction rates of DMSe, impact Se deposition in polar regions. Slower reaction rates of DMSe with OH ($k_6$) and $O_3$ ($k_8$) enhance deposition over the example

Antarctic grid box (Fig. 10a-c). Longer DMSe lifetimes allow more marine Se emissions to reach polar regions, which have little local Se emission sources. The chemical rate coefficients are more important in the Antarctic than the Arctic since there



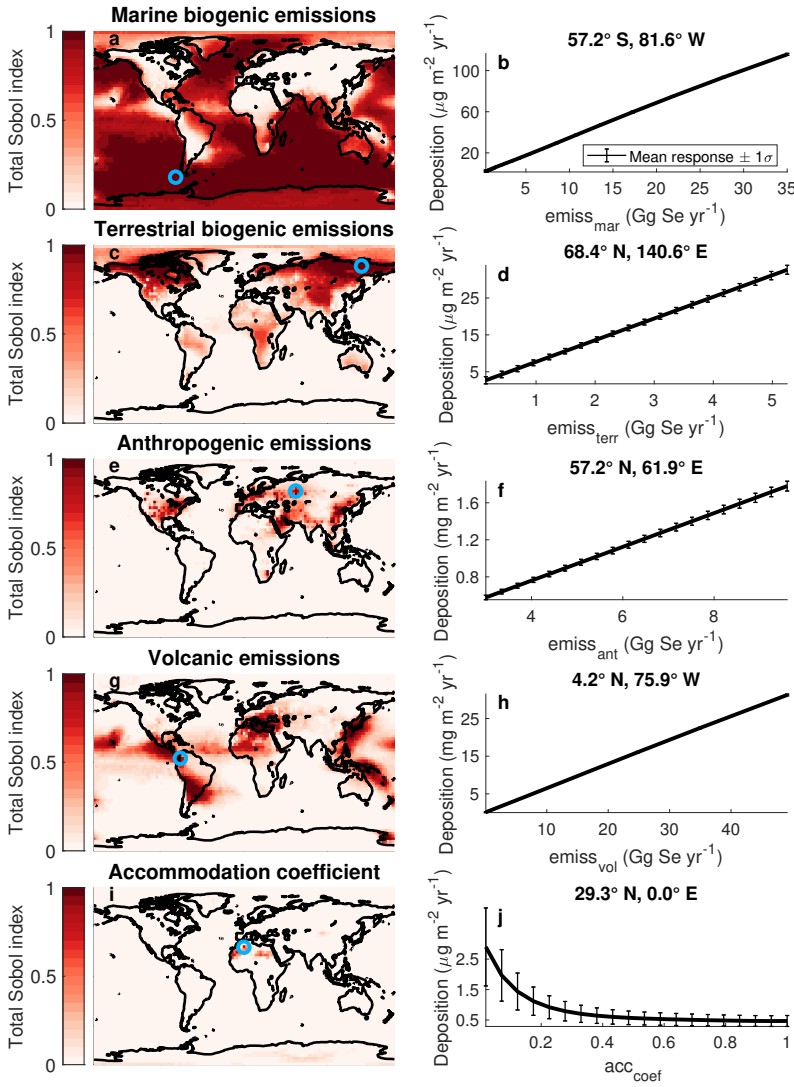

**Figure 9.** Maps of the total Sobol' indices of emission parameters and the accommodation coefficient for total Se deposition (*left column*). A blue circle indicates the grid box where the total Sobol' index is maximum. The relationship between total Se deposition and the input parameters in that grid box, calculated by resampling the surrogate model for deposition (*right column*). Note that the magnitude of deposition (*y*-axis) varies in each plot, depending on the grid box shown.

is more $O_3$ and OH in the Northern Hemisphere than the Southern Hemisphere, meaning that the DMSe lifetime is longer in the Southern Hemisphere than the Northern Hemisphere. Dummy aerosol parameters are only important for Se deposition in the Arctic (Fig. 10d–f). We calculate the relationship between Se deposition and dummy aerosols using the same quantity for emitted surface area of the dummy aerosols as in Fig. 7c. With increasing dummy aerosol surface area, the Se deposition in 5 the Arctic increases, but only after surpassing the threshold of the available sulfate aerosol surface area in that latitude band





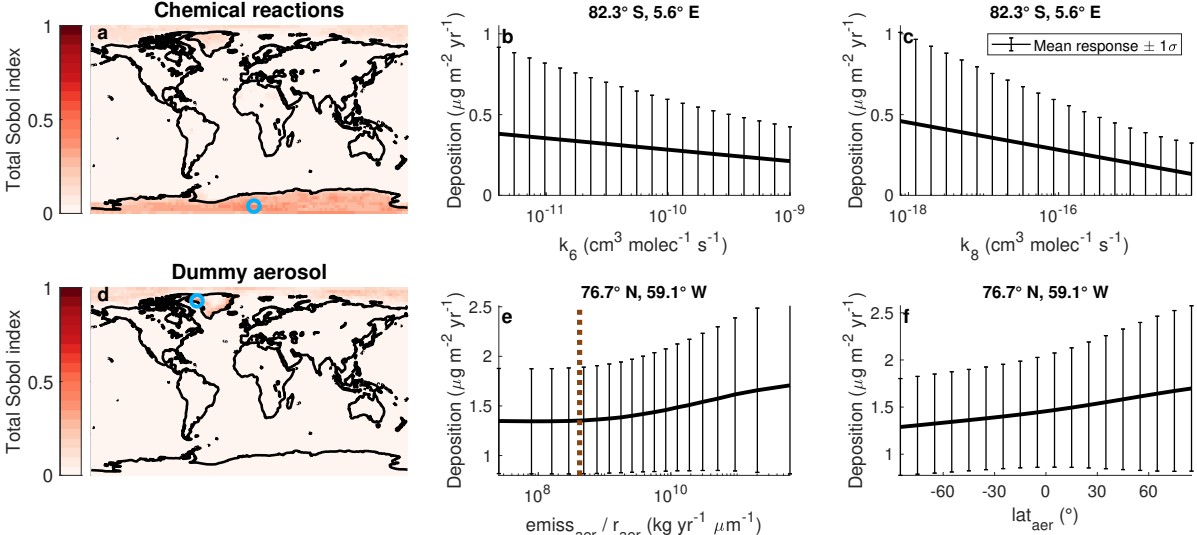

**Figure 10.** Maps of the total Sobol' indices of reaction rate constants and dummy aerosol parameters for total Se deposition (*left column*). A blue circle indicates the grid box where the aggregated total Sobol' index is maximum. The relationship between total Se deposition and the multiple relevant input parameters within the aggregated index for that grid box is shown (*center and right columns*). The emitted dummy aerosol surface area is compared to the corresponding sulfate quantity in the latitude band of the grid box, shown as a dashed brown line in (*e*).

(Fig. 10e). The dummy aerosols also have a stronger effect on Se deposition when they are emitted in a latitude band closer to the example grid box at 76.7° N (Fig. 10f). Attachment of oxidized Se to dummy aerosols increases the overall lifetime of Se (Sect. 5.1), leading to enhanced transport of Se to the Arctic region. The transport of Se on dummy aerosols does not lead to higher deposition in the Antarctic, perhaps because wet deposition in the Antarctic circumpolar storm track impedes the

5 transport of aerosol poleward.

It is also important to note which input parameters do not influence Se deposition in any of the grid boxes. Variations in the speciation of emissions, photolysis rates, and 15 of the Se reaction rate constants have a negligible influence on deposition.

Although other parameters may play a role in certain grid boxes, the emission parameters are most important on the global scale, evidenced by their higher mean total Sobol' index (Fig. 11). Figure 9 illustrates which regions are affected by different

emission sources. Variations of marine emissions impact the most grid boxes; however, their influence is mainly confined to the oceans, coastal areas, and Southern Hemispheric continents. Since the motivation of studying Se deposition is to understand its impact on agricultural soils, we also calculated the mean influence of parameters in pasture and cropland areas (Fig. 11), using maps from Ramankutty et al. (2008). The importance of anthropogenic emissions increases when looking only at pasture and cropland areas, since agricultural areas coincide with human settlement. All four emission parameters show similar levels

of importance for agricultural regions. Therefore, further work in understanding any of these emission processes would be valuable to reducing the uncertainty in deposition fluxes.





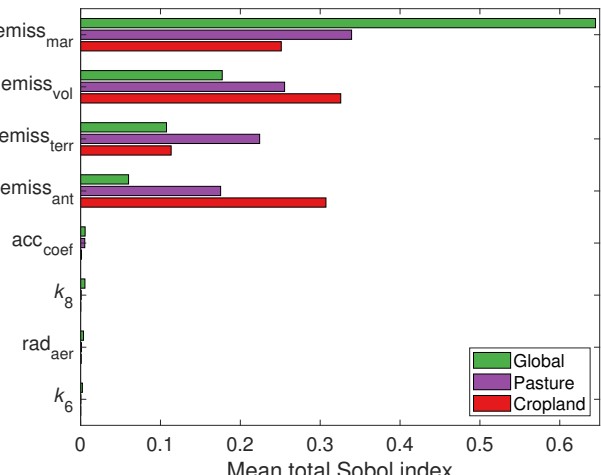

**Figure 11.** Bar plot summarizing the importance of the input parameters to total Se deposition globally and in agricultural areas. For the cropland and pasture means, Sobol' indices are averaged over grid boxes that are covered by more than 25% cropland or pasture area in the Ramankutty et al. (2008) database.

## 5.3 Comparison with deposition flux measurements

With surrogate models of wet Se deposition, we can estimate the modeled wet Se deposition throughout the parametric uncertainty space. By comparing these modeled distributions of wet deposition with observed values, one could constrain input parameters to which deposition is sensitive. However, we do not attempt at this stage to calibrate the parameters to existing

measurements because of several challenges in comparing the compiled measurement dataset with the simulations in this study. Firstly, the emissions and meteorology in this study are representative of the year 2000, whereas the measurements were made between 1975 and 2017. Secondly, Se is a difficult element to measure at environmental concentrations, which might lead to inaccurate reported deposition fluxes. The most popular analytical method is inductively coupled plasma mass spectrometry (ICP-MS), which was used for 58 of the 73 sites in the database. However, it is difficult to measure Se with ICP-MS due

to the low ionization of Se, the Se signal being split on the five stable isotopes, and especially mass interferences (Winkel et al., 2012). Several studies reported that Se concentrations in rainwater samples were often below the detection limit of the analytical method (e.g., Arimoto et al., 1987; Gratz et al., 2013). Unfortunately, other studies often do not explicitly report the detection limit, the fraction of samples under the detection limit, and how these samples are treated statistically. Thirdly, many of the measurement sites were located in urban locations close to point-source emissions. Due its coarse resolution ($2.8° \times$

$2.8°$), the model would have difficulty reproducing point values for Se concentrations and deposition fluxes.

Figure 12 compares the measured Se wet deposition fluxes with the resampled median deposition fluxes from the surrogate models, also showing the likely range predicted by the models (defined with bounds of the 2nd and 98th percentile values). With the results from the deposition sensitivity analysis (Fig. 9), we categorize each measurement location by the input pa-





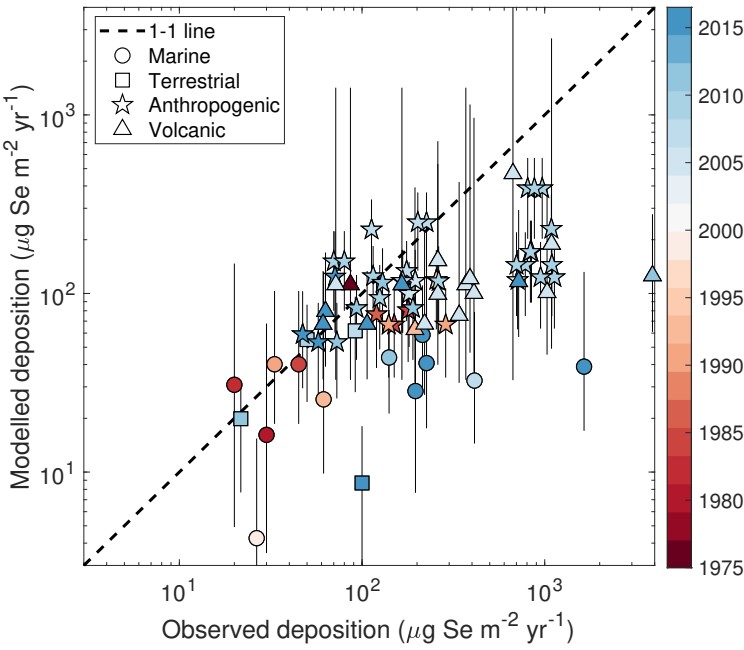

**Figure 12.** Comparison of wet deposition flux measurements (Table 5) with modeled fluxes. Medians for the modeled values are shown, along with vertical bars representing the $2^{nd}$ and $98^{th}$ percentile values. The symbols correspond to the model input parameter that is most important for deposition at the measurement location. The color in the points represents the year when the measurement was taken; multi-year measurements show the middle year.

rameter that predominates the uncertainty in modeled deposition, indicated by the symbol in Fig. 12. The compiled data and measurements show good agreement at the lower end of deposition values, where the measurement sites are more remote from point-source emissions. The agreement worsens at higher values of observed Se deposition, which correspond to more urban measurement sites. As discussed before, it is not surprising that the model has difficulty matching the measurements for

5   higher observed values of deposition, since the model is coarse resolution and the simulation year may be mismatched from the measurement year. Indeed, the model underestimates several Se deposition measurements in urban sites from East Asia after 2005. Anthropogenic $SO_2$ emissions, an analogue of anthropogenic Se emissions, has increased in East Asia since 2000 (Smith et al., 2011). Natural emission factors dominate the variability in several of these locations in the year 2000 simulations, likely because the input anthropogenic emission maps do not correspond to the measurement time period. Nevertheless, we find

10   overall that 53% of existing measurements are within the likely range of the model's prediction. The agreement improves to 79% when comparing the model range to measurements in background locations, defined as having observed deposition values below $150\,\mu g\,Se\,m^{-2}$. These results provide confidence to SOCOL-AER's predictions of Se deposition fluxes in non-urban locations.





## 6 Discussion

Through our consideration of model uncertainties related to Se cycling, we derived a median atmospheric Se lifetime of 4.4
d. This is the first bottom-up estimate for Se made with a mechanistic global atmospheric model. Our estimate for the Se
lifetime matches the global arsenic lifetime calculated in the global model of Wai et al. (2016), namely 4.5 d. This agreement
is due to both elements attaching to submicrometer aerosol particles; therefore, their lifetime is determined by the lifetime
of these particles. Since Se and arsenic have similar atmospheric sources as well (e.g., metal smelting, coal combustion,
volcanoes), it may be possible to draw analogies between their atmospheric cycles. The range of previously estimated Se
lifetimes from global atmospheric budgets is between 0.8 and 6 d, similar to our result (Ross, 1985; Mosher and Duce, 1987).
The recent value from Mason et al. (2018) of a 0.15 yr (55 d) Se lifetime seems overestimated compared to our results and
past budgets, especially since Mason et al. (2018) only consider gas-phase Se in their model, which tends to be shorter lived
in the atmosphere than aerosol-bound Se. According to our sensitivity analysis results, the atmospheric Se lifetime could be
further constrained by measuring the OCSe + OH reaction rates, and in general knowing more about whether OCSe is present
in the atmosphere. Since dummy aerosols also impact the Se lifetime in our model, implementing a more complex tropospheric
aerosol parametrization in SOCOL-AER would also further constrain the atmospheric lifetime of Se. However, since the main
interest in Se is its atmospheric input to agricultural soils, it may be a higher priority to constrain the input parameters that
affect the deposition of Se in agricultural regions rather than the Se lifetime

The results of the sensitivity analyses raise an obvious question: why do the input parameters that influence the atmospheric
Se lifetime not appear as important for the Se deposition fluxes? One would expect that Se deposition fluxes close to areas of
high emissions would be dominated by the magnitude of these emissions. One would also expect that, if anywhere, variation in
the Se lifetime would play a role over remote regions, where the amount of locally emitted Se is low and thus the amount that
can be transported from emission regions has a larger effect on deposition. However, the range in the atmospheric Se lifetime
in our simulations is relatively narrow, between 2.9 and 6.4 d if we consider the 2[nd] percentile and 98[th] percentile bounds (Fig.
4). On the other hand, emissions of various Se species can vary by orders of magnitude (Table 3). These larger variations in
the amount of emitted Se have a larger impact on deposition than smaller variations in the Se lifetime, even in many remote
places. Only in extremely remote areas, for example in the Arctic, do some of the parameters that affect the Se lifetime show
up as important, like the dummy aerosols. Parameters with regional rather than global importance for the Se lifetime, like the
DMSe reaction parameters, impact deposition of Se in the Antarctic by controlling the amount of transported Se. It is not
surprising that the parameter that has the largest impact on lifetime, the OCSe + OH reaction rate constant, has little impact on
deposition fluxes, since emissions of OCSe are assumed to be a minor flux of Se (maximum 6% of the anthropogenic emissions
flux). Like all sensitivity analyses, the results are dependent on the choice of uncertainty ranges for the different parameters;
if we had selected narrower uncertainties for the Se emission sources, the uncertainties of parameters that affect Se lifetime
(e.g., chemical reaction rates, dummy aerosols, etc.) may have been more important in remote regions. However, the choice of
wide uncertainty ranges for the Se emissions is justified, given the variability in natural emission processes and a lack of field





campaigns assessing Se emission fluxes (Sect. 3.1.4). The different results for the two types of sensitivity analyses (lifetime and deposition fluxes) highlight that the "important" parameters to constrain depend on the choice of research question.

The global sensitivity analyses in this paper provides clear next steps for atmospheric Se research. The magnitude and spatial distribution of Se emissions remain the most important uncertainty to constrain, in order to improve the predictions of

Se deposition patterns. Further investigations of chemical reactivity of Se species or the speciation of emissions are a lower priority, although measuring the speciation of emissions can give mechanistic insights into emission processes. The emission uncertainties could be constrained by conducting field campaigns that either measure emission fluxes of Se close to sources (e.g., Amouroux et al., 2001) or separate Se source contributions at an ambient measurement site through trajectory modeling and/or speciation measurements (e.g., Suess et al., 2019). Our model results can help identify ambient locations that would

be interesting to study for field campaigns, by mapping the contribution of the Se emission sources to deposition in different regions (Fig. 9). In addition to new field measurements, we can also compile and reanalyze previously collected data from the literature to evaluate estimates of emission fluxes. Bayesian inverse modeling techniques (e.g., Stohl et al., 2009) could be employed in conjunction with the SOCOL-AER model to provide posterior estimates for Se emission fluxes. Global sensitivity analysis is an invaluable first step before such model calibration techniques, since the parameter dimensionality can be reduced

by neglecting non-influential parameters. As shown in Sect. 5.3, the heterogeneity of compiled literature data represents a challenge to comparing models and measurements. Therefore, standardized measurement techniques and adequate reporting of sampling, analytical, and post-processing methods are required so that the model is not calibrated to an errant measurement.

## 7   Conclusions

Now that it includes Se cycling, the SOCOL-AER model can be used to predict Se transport and deposition globally. We

created surrogate PCE-based models that are able to predict the output of the model throughout the uncertainty space of the input parameters. With these surrogate models, we determined that the atmospheric Se lifetime is around 4.4 d, similar to the lifetime of submicron aerosol particles in the atmosphere. Assuming that longitudinal wind speeds are around 10 m s$^{-1}$ (Jacob, 1999), the likely Se lifetime range of 2.9–6.4 d corresponds to a distance of 2500–5000 km that Se is transported in the atmosphere. The global sensitivity analysis of Se deposition fluxes shows that reducing uncertainties in Se emissions would

lead to the biggest reductions in the uncertainty of deposition maps. Field measurements that elucidate and quantify Se emission processes should be prioritized, so that model predictions of Se deposition maps can be improved. Available measurements of Se in rainwater are within the likely range of model results at 79% of background sites; remaining discrepancies may be due to the time period of the simulations in this study, the coarse resolution of the model, and analytical challenges leading to measurement inaccuracies. In a future study, SOCOL-AER can be applied to different time periods to investigate how Se

deposition has changed due to variations in anthropogenic emissions and climate.



*Code and data availability.* The SOCOL-AER code is available upon request from the authors, after users have signed the ECHAM5 license agreement http://www.mpimet.mpg.de/en/science/models/license/. The relevant simulation data, along with the experimental design of the training runs, is available at: https://doi.org/10.3929/ethz-b-000357105 (Feinberg et al., 2019a). The compiled Se precipitation database is available in the Supplement. The NAtChem precipitation database is available online at: http://donnees.ec.gc.ca/data/air/monitor/

monitoring-of-atmospheric-precipitation-chemistry/metals-in-precipitation/. Selenium in precipitation measured by EMEP was extracted from annual reports on heavy metals and POP measurements available at: https://projects.nilu.no//ccc/reports.html. UQLAB is freely available for Academic and degree-granting institutions by registering at: https://www.uqlab.com/register. All of the scientific source code is available under the BSD 3-clause license and can be downloaded from: https://www.uqlab.com/obtain-the-sources.

*Author contributions.* AS, TP, and LW initiated the project of studying the atmospheric Se cycle using a chemistry–climate model. AF

implemented Se into SOCOL-AER with assistance from AS, conducted all simulations, analyzed the results, and wrote the paper with contributions from all authors during the revision process. AF was directly supervised by AS, TP, and LW during the project. MM and BS guided the statistical analysis during the project. MM developed the scripts to conduct the global sensitivity analysis on the Se model and aided with writing the statistical methods section.

*Competing interests.* The authors declare that they have no conflict of interest.

*Acknowledgements.* This work was supported by a grant from ETH Zürich under the project ETH-39 15-2. Thanks to S. Gysin for his work on sensitivity analysis in a Se box model during his bachelor thesis at ETH. Thanks to D. Amouroux and E. Tessier from CNRS/University of Pau for helpful discussions about marine DMSe emissions. Thanks to F. Genter for her help in compiling past studies for the Se precipitation database. We acknowledge Environment and Climate Change Canada's Monitoring and Surveillance in the Great Lakes Basin (GLB) under the Chemicals Management Plan (CMP) for the provision of Se in precipitation data, and specifically H. Dryfhout-Clark for providing

additional information about this dataset. We acknowledge B. Trost and A. Breuninger for providing data from the Alaska Department of Environmental Conservation. Thanks to A. Meharg for providing additional data of Se in precipitation. We acknowledge all scientists who contributed to measurements that were compiled into the Se precipitation datasets. Thanks to the developers of the UQLAB for providing the software used in this study. Thanks to T. Meschini for the design of the Se cycle schematic. Color tables from ColorBrewer 2.0 (www.colorbrewer2.org) were used for the figures in this paper.



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
