# Peer review of "Mapping the drivers of uncertainty in atmospheric selenium deposition with global sensitivity analysis"

_Atmospheric Chemistry and Physics, 2019_

## Referee Comment (RC1) · Anonymous Referee #1 · 10 Oct 2019

General comments:

This paper represents an excellent uncertainty analysis of simulating the atmospheric chemistry and deposition of Se species. The uncertainty analysis is thorough, fully described, and discussed in detail. The authors recommend, based on this analysis, that the highest leverage point in uncertainty reduction in future work is the quantification of emissions of Se because the output of highest interest is deposition to agricultural soils. This is a relevant conclusion, and it is supported by the results. Overall, the quality of this paper is high and it is satisfactory in all aspects.

Specific questions:

[Figure]

While it is likely not a factor for the main conclusions, the boundary conditions for the model represent the year 2000. How might inter-annual differences or recent trends in these assumed conditions propagate to outputs like atmospheric Se lifetime? For example, via particle abundance, meteorology, etc.

In the absence of better information, the authors scale Se emissions spatially using the spatial distribution of S emissions. This is certainly a reasonable starting point. Given the large uncertainties in emissions magnitudes, how would the spatial uncertainty in S:Se ratio compare? The authors mention that the magnitude and spatial distribution of emissions are the most important uncertainties to constrain, but how do those two compare to each other?

The reactions modeled are assumed to have no temperature dependence (rates for 298K used). Most of the atmosphere, however, is significantly colder than this. How would the unquantified temperature dependence affect the given results?

The range of photolysis frequencies are scaled from 0-2 times with a uniform distribution. Photolysis frequencies in general vary substantially between chemical species; why are these rates treated on a relatively narrower range than the reaction rates, which do vary over orders of magnitude?

Technical corrections:

None

---

## Referee Comment (RC2) · Anonymous Referee #2 · 13 Oct 2019

This paper introduces a newly developed global model for simulating the atmospheric cycling of selenium and also presents a detailed uncertainty analyses for the model simulations. I find this to be a very interesting study and I can see lots of efforts has been put into this work. A few specific questions and comments I have –

1) It is desirable to report the Se emissions calculated in the model, preferably in a table. Since there have been some literature on this (e.g. Mosher and Duce, 1987 and some more recent ones) which clearly summarized the Se emissions from all major sources (including volcanic, marine biosphere, terrestrial biosphere, and anthropogenic sources), it would be very helpful to compare results from this study with the literature.

[Figure]
* * *
Interactive
comment

2) It is reported that the model tracks Se in 7 gas-phase species and 41 aerosol tracers, it would be helpful to clarify how much are emissions for each of the Se species.

3). "all four Se sources (volcanic, marine biosphere, terrestrial biosphere, and anthropogenic emissions) contributing equally to the uncertainty in deposition over agricultural areas." – I'm kind of surprised with this and I don't see enough data supporting this conclusion.

4). Section 5.1, the discussion on atmospheric lifetime of Se – it's useful to look at the parameters related to emissions, but the more important factor regulating the atm. Lifetime is meteorology, in particular precipitation (including both the precipitation amount and spatial distribution). Is the year 2000 happen to be a dry or wet year (indicated by the meteorology data you use for that year)? How much would be the likely variation of atm. lifetime caused by the interannual variability in meteorology? More discussion on this would be more relevant and valuable.

---

## Referee Comment (RC3) · Anonymous Referee #3 · 17 Oct 2019

The authors carry out a sensitivity analysis of atmospheric selenium deposition using polynomial chaos expansion as a surrogate model. In general, this is a new and interesting application of surrogate modelling and sensitivity analysis and so has the potential to represent a forward step in the field. Additional information is required on the methods carried out to convince me that the surrogate model is suitable for the sensitivity analysis, particularly for Se Lifetime.

In particular: 1. There are some choices made in the application of the polynomial chaos expansion that are not explained and it's difficult to know whether they are rule of thumb decisions or made specifically for this application. How was q=0.75 decided

upon and what other choices were there to least-angle regression?

2. I don't follow the logic of explaining LOO validation and then using a model that doesn't actually leave any of the training runs out. Can you explain why this is appropriate and how the validation might change if you had used LOO? Is it appropriate to continue to call it LOO validation with a full model? It may be explained in one of the given references but I'd like to see some explanation here.

3. You choose the degree based on whether the validation doesn't decrease in the next step. Why and how does the validation increase? I would expect the validation to always decrease to some extent with extra terms. If a threshold is applied, what is it?

4. It seems counter-intuitive that calculating the surrogate separately for burden and deposition yields better results for lifetime than directly modelling lifetime? I would expect there to be double the errors. How was 'better' calculated and can you explain a bit more why it's a fair result?

5. Can you explain a little more about how the sensitivity analysis is derived from Equation 4? It's not enough here to refer to previous work.

6. Given the results in Figure 5 I'm not convinced that double counting interactions because they are less than 0.05 is a good idea. 0.05 is quite a large fraction of the 0.15 that is the largest main effect. Can you find a way to investigate the effect this is having?

7. In Figure 5, the main effects are quite low and nowhere near adding up to 1 – was the amount of interaction in this model expected?

8. Still with Figure 5, in previous experience seeing interactions that are large and consistent between multiple variables is a sign that the model fit is actually poor. It's not clear because of the way you have carried out LOO and added the interaction terms whether this is indicating poor model fit or whether these are real interactions.

9. The main effect figures show that there is not much range on the y-axis covered by

the central line – it's highlighted by the uncertainty in the remaining parameters. Could you add some information on how much uncertainty there is from using PCE as your surrogate? I would like to see this to show me that most of what you are seeing is not simply a result of the use of a surrogate model.

---

## Author Comment (AC1) · 20 Nov 2019

**Author response to all referees' comments on "Mapping the drivers of uncertainty in atmospheric selenium deposition with global sensitivity analysis"**

We would like to thank the three reviewers for their time and helpful comments about the manuscript. We appreciate their interest in the manuscript and think that they have raised pertinent points about our analysis. We have considered these comments carefully and present our responses below, as well as changes to the main text (reviewer comment are in blue and author responses are in black).

**Response to Referee #1**

While it is likely not a factor for the main conclusions, the boundary conditions for the model represent the year 2000. How might inter-annual differences or recent trends in these assumed conditions propagate to outputs like atmospheric Se lifetime? For example, via particle abundance, meteorology, etc.

We agree with the reviewer's point that this is important to consider. Since we have only tested the sensitivity of model parameters in a certain year (2000), we are not able to comment on how the meteorology of different years would affect the atmospheric Se lifetime. We believe that the largest interannual change to consider is precipitation shifts and their effect on Se deposition, 80% of which is through wet deposition. For example, there was a La Niña event in the year 2000 (`https://origin.cpc.ncep.noaa.gov/products/analysis_monitoring/ensostuff/ONI_v5.php`), which would cause higher precipitation than normal over the Western Pacific and various teleconnections. We restricted the scope of our sensitivity analysis to Se cycle input parameters, which already cover a 34-dimensional space. To avoid computational expense, we did not investigate the dependencies of model output on climate-related parameters in this sensitivity analysis. However, we intend to consider the effects of inter-annual variability and long-term climate change in future studies. We added this caveat to the discussion, P32L10–12:

"It must be noted that our simulations were performed only for the year 2000 and focused on uncertainties in the Se-related input parameters, neglecting variations in the Se lifetime and deposition due to interannual variability in meteorology and sulfate aerosol properties. In future studies we intend to investigate how the Se cycle varies under different climate conditions."

In the absence of better information, the authors scale Se emissions spatially using the spatial distribution of S emissions. This is certainly a reasonable starting point. Given the large uncertainties in emissions magnitudes, how would the spatial uncertainty in S:Se ratio compare? The authors mention that the magnitude and spatial distribution of emissions are the most important uncertainties to constrain, but how do those two compare to each other?

This is a very good point. We followed a similar thought process as the reviewer, and considered this assumption to be a reasonable starting point for our sensitivity analysis. There are likely spatial uncertainties in the S:Se ratio, however the number of parameters we could consider in our sensitivity analysis was limited due to computational expense. Since the results showed that emissions are the most important to constrain for predicting Se deposition maps, we intend to investigate these spatial uncertainties in future studies. For natural sources, this is very difficult, due to a lack of studies. For example, there

is only one oceanic field study that has concurrently measured both S and Se fluxes (Amouroux et al., 2001). A current study is underway in our group to measure marine Se and S fluxes in various locations, which will improve our knowledge about these processes. In terms of anthropogenic variations of Se:S ratios, Lee et al. (2015) suggest that although the Se content of Chinese coal is double that of U.S. coal, S varies in a similar proportion and thus the Se:S ratios of both types of coal are similar. Nevertheless, there needs to be further research into whether regionally varying coal cleaning methods and pollution control technologies can alter the Se:S ratio of emissions.

The reactions modeled are assumed to have no temperature dependence (rates for 298K used). Most of the atmosphere, however, is significantly colder than this. How would the unquantified temperature dependence affect the given results?

As mentioned in the paper, we could not find any studies that measured the temperature dependence of the Se reaction rates, and therefore did not include this in the sensitivity analysis. We looked at the temperature dependence of the analogous S reactions (Burkholder et al., 2015) between 298 K and 240 K (covering most conditions of the lower troposphere, where we expect gaseous Se species to occur). Between these temperatures, the DMS + OH reaction rate decreases by 20%, DMS + $NO_3$ increases by 54%, $H_2S$ + OH decreases by 6%, and OCS + OH decreases by 58%. These changes are all smaller than the uncertainty range that we assume for these reaction rates, spanning orders of magnitude. Therefore, we do not think that the unquantified temperature dependence would have a strong effect on our sensitivity analysis.

The range of photolysis frequencies are scaled from 0-2 times with a uniform distribution. Photolysis frequencies in general vary substantially between chemical species; why are these rates treated on a relatively narrower range than the reaction rates, which do vary over orders of magnitude?

As we mention in the paper, there is a lack of information about the uncertainties involved in the photolysis rates. Since the quantum yield of these photolysis reactions could theoretically be 0, we set the lower bound of the photolysis scaling factor as 0. To yield a range that is symmetrical around a unity scaling factor, we set the upper bound as 2. Since there is generally very little sensitivity of deposition fluxes to reaction rates (which we varied by orders of magnitude), we do not think that using a wider range for photolysis rates would cause these parameters to become influential.

**Response to Referee #2**

It is desirable to report the Se emissions calculated in the model, preferably in a table. Since there have been some literature on this (e.g. Mosher and Duce, 1987 and some more recent ones) which clearly summarized the Se emissions from all major sources (including volcanic, marine biosphere, terrestrial biosphere, and anthropogenic sources), it would be very helpful to compare results from this study with the literature.

Thank you for your comment. The ranges used in this sensitivity analysis for Se emissions from different sources were listed in Table 3. We have now included a summary of past studies estimates in the supplementary material (Table S1).

It is reported that the model tracks Se in 7 gas-phase species and 41 aerosol tracers, it would be helpful to clarify how much are emissions for each of the Se species.

The sources of each Se species are listed in Table 1. We vary the amount of emissions for each species in the sensitivity analysis, between the ranges shown in Table 3. Therefore, in each simulation there can be a different amount of each species emitted.

"all four Se sources (volcanic, marine biosphere, terrestrial biosphere, and anthropogenic emissions) contributing equally to the uncertainty in deposition over agricultural areas. I'm kind of surprised with this and I don't see enough data supporting this conclusion.

We base this conclusion on Figure 11, which shows the total Sobol' index averaged over different land types. Over the entire globe, marine emissions have the largest influence (green bars), due to oceans covering 70% of the Earth. However, if we only average over pasture (purple bars) or cropland (red bars) areas, then all emission sources show similar mean Sobol' indices between 0.15–0.30. We now clarify this on P28L15:

"All four emission parameters show similar levels of importance for agricultural regions, i.e. showing similar mean Sobol indices for pasture and cropland areas."

Section 5.1, the discussion on atmospheric lifetime of Se it's useful to look at the parameters related to emissions, but the more important factor regulating the atm. Lifetime is meteorology, in particular precipitation (including both the precipitation amount and spatial distribution). Is the year 2000 happen to be a dry or wet year (indicated by the meteorology data you use for that year)? How much would be the likely variation of atm. lifetime caused by the interannual variability in meteorology? More discussion on this would be more relevant and valuable.

This point was brought up by Referee #1 as well, which we addressed above. We have now included in the discussion a caveat that our sensitivity analyses do not cover variations of the atmospheric Se lifetime due to interannual variability in climate and emissions.

**Response to Referee #3**

Additional information is required on the methods carried out to convince me that the surrogate model is suitable for the sensitivity analysis, particularly for Se Lifetime.

Thank you for your feedback, which made us consider again how we validated the surrogate models. We decided to run an additional validation set using the full SOCOL-AER model, to verify the suitability of our surrogate models. We chose the parameters for the 50 additional validation runs by enriching the existing training set so that a pseudo-Latin hypercube of 450 runs was formed (using the UQLAB function uq_LHSify). This ensures that our validation set is testing new regions of the parameter space compared to the training set.

In the figure below, we show the results of the validation runs compared to the predictions of the surrogate models for global Se burden, deposition flux, and lifetime. The compared values are shown as blue circles and the 1:1 line as a red line. The global lifetime surrogate model is calculated by dividing the PCE of the global burden by the PCE of the global deposition flux. We had found the LOO error of the burden PCE to be around 0.02 and the LOO of the deposition flux PCE to be on the order of $10^{-6}$. We can compare these LOO errors to the validation errors, which are calculated from the independent validation set (Marelli and Sudret, 2019):

$$\text{Error}_{val} = \frac{N-1}{N} \left[ \frac{\sum\limits_{i=1}^{N} \left( \mathcal{M}(\mathbf{x}_{val}^{(i)}) - \mathcal{M}^{PCE}(\mathbf{x}_{val}^{(i)}) \right)^2}{\sum\limits_{i=1}^{N} \left( \mathcal{M}(\mathbf{x}_{val}^{(i)}) - \hat{\mu}_{Y_{val}} \right)^2} \right]$$

where $N$ is the number of validation runs, $\mathcal{M}(\mathbf{x}_{val})$ is the SOCOL-AER model output of the validation runs, $\mathcal{M}^{PCE}(\mathbf{x}_{val})$ is the prediction of the surrogate model for the sample points in the validation set, and $\hat{\mu}_{Y_{val}}$ is the sample mean of validation set output.

[Figure]

The validation errors match the LOO errors for global Se burden and Se deposition, showing that the LOO error is behaving correctly. However, as the referee suspected, the surrogate model for the global lifetime shows a relatively high validation error of around 0.35. The global selenium lifetime is clearly a

difficult parameter to emulate, likely requiring much more than 400 training runs to be accurately modelled. However, we think that the general results of the lifetime sensitivity analysis would not change, especially the identification of the most important parameters (OCSe reaction rates and dummy aerosols). We still think that the discussion of the selenium lifetime sensitivity analysis is valuable in the paper, especially as an introduction to the reader about a sensitivity analysis with one model output.

In the second sensitivity analysis of the paper, we looked into the factors that affect Se deposition in each grid box. The independent validation dataset verifies the accuracy of the deposition flux surrogate models. We compare the simulated deposition in the 50 validation runs at all 8192 horizontal grid boxes with the PCE-predicted values, i.e. $50 \times 8192 = 409\,600$ points. The left plot shows the results in linear space and the right plot in logarithmic space, since in the linear plot the smaller deposition fluxes collapse around 0. Most points fall around the 1:1 line, with the validation error being 0.013. Therefore, these surrogate models have much better accuracy than the lifetime surrogate model and are not affected by the issues brought up by the reviewer.

[Figure]

We have added these results to the supplementary information of the paper. In the main paper, we have added in the methods section how we produced the validation dataset (P18L9–13):

"The cross-validation approach would usually remove the need for an independent validation dataset, saving computational expense. However, to evaluate the post-processing steps applied to the surrogate models, we also produced an independent validation dataset of 50 SOCOL-AER runs. The parameters for these runs were chosen by enriching the training experimental design to create a pseudo-Latin Hypercube of 450 runs, ensuring that the distance between the validation runs and existing training runs is maximal."

We also now acknowledge the accuracy of the lifetime surrogate model compared to the validation dataset in the main paper (P21L24–26):

"We derive a surrogate model for the atmospheric Se lifetime by dividing the Se burden PCE model by the Se deposition PCE model (Sect. 3.3). This surrogate model shows a higher error (0.35) than the burden and deposition flux PCE models, which would only be reduced by running more training runs (Fig. S2)."

There are some choices made in the application of the polynomial chaos expansion that are not explained and it's difficult to know whether they are rule of thumb decisions or made specifically for this application. How was q=0.75 decided upon and what other choices were there to least-angle regression?

The choice of q=0.75 is arbitrary and based on our own experience. The range [0.7-0.8] seems to be a good compromise between computational expense and accuracy of the PCE model. We now rephrase this on P16L27:

"Based on previous experience, we selected a $q$ value of 0.75".

I don't follow the logic of explaining LOO validation and then using a model that doesn't actually leave any of the training runs out. Can you explain why this is appropriate and how the validation might change if you had used LOO? Is it appropriate to continue to call it LOO validation with a full model? It may be explained in one of the given references but I'd like to see some explanation here.

To calculate the LOO error, the process of leaving out one training point and calculating a new PCE model is repeated $N$ times, where $N$ is the size of the training dataset. At each stage of the LOO error calculation, the same basis set of polynomials is used to calculate all PCEs. The final basis set that is selected by the algorithm is the one that minimizes the LOO error, ensuring that the PCE calculated with the full training set does not overfit the data. Previous studies have found that the LOO error performs well in predicting the generalization error of a statistical model (Molinaro et al., 2005; Blatman and Sudret, 2010), which we also found when comparing the LOO errors to our independent validation set errors. The LOO error for a PCE model would be calculated using Eq. 7 (in the paper), which requires the calculation of $N$ PCE models. An equivalent form of Eq. 7 is used (Eq. 8), which requires the calculation of a single PCE model.

You choose the degree based on whether the validation doesn't decrease in the next step. Why and how does the validation increase? I would expect the validation to always decrease to some extent with extra terms. If a threshold is applied, what is it?

Technically, the error should decrease as the degree increases. However, in practice the sample size is given and does not change as we increase the degree. The number of coefficients to estimate does increase together with the polynomial degree but in the process we do not bring any new information (no additional training points). Therefore, it becomes increasingly difficult to estimate the coefficients with the limited available information. On top of that, the least-square solution involves inverting a matrix which leads to a larger numerical error as the matrix size increases. For these reasons, the error starts to increase at a certain degree. The algorithm implemented in UQLab is aware of these issues and thus stops the algorithm when it realizes that the error is not decreasing despite increasing the degree for two successive iterations.

It seems counter-intuitive that calculating the surrogate separately for burden and deposition yields better results for lifetime than directly modelling lifetime? I would expect there to be double the errors. How was 'better' calculated and can you explain a bit more why it's a fair result?

The output of interest, the atmospheric Se lifetime, is of the form $Y = \frac{a}{b}$, where $a$ is the global Se burden and $b$ is the deposition flux. If $a$ and $b$ are linear or can easily be approximated by polynomials it is better to do so than trying to approximate a function Y, which is possibly of degree order $-1, -2$, etc. (which would eventually be possible but would require a larger experimental design).

We can confirm this approach with our new independent validation set. We produced a PCE of the Se lifetime directly, and compared this PCE's predictions with the 50 validation runs (figure shown below). The validation error, 0.71, is worse than the validation error we found when computing PCEs for $a$ and $b$ separately and calculating the lifetime from these two PCEs, which had a validation error 0.35 (compare to lower right figure on page 4). Therefore, this approach of calculating separate surrogate models for burden and deposition succeeded in improving the accuracy of the surrogate lifetime model.

[Figure]

Can you explain a little more about how the sensitivity analysis is derived from Equation 4? It's not enough here to refer to previous work.

We decided to include a general description of how the sensitivity indices can be derived from the PCE, while for a more mathematical description the reader can refer to the Sudret (2008) paper. We now include on P20L5–9:

"As shown by Sudret (2008), Sobol sensitivity indices are a function of the calculated coefficients in the PCE model (Eq. 4), due to the similarity of the PCE decomposition with variance decomposition. First order Sobol' indices can be calculated as $D_i = \sum_{\alpha \in \mathcal{A}_i} y_\alpha^2$, where $\mathcal{A}_i$ is the set of polynomial terms involving only variable $i$. Similarly, higher-order Sobol' indices can be calculated by summing the squares of the coefficients of the polynomial terms that include the variables of interest."

Given the results in Figure 5 I'm not convinced that double counting interactions because they are less than 0.05 is a good idea. 0.05 is quite a large fraction of the 0.15 that is the largest main effect. Can you find a way to investigate the effect this is having?

As we acknowledge in the paper (P20L16–17), it is an upper limit for the overall dummy aerosol effect. In order to calculate the effect this would have, we would have to sum the individual Sobol' indices. We are limited in this by numerical issues when the Sobol' indices are calculated by Monte Carlo methods. However, we can calculate this when the surrogate model is a PCE, since the Sobol' indices can be calculated analytically. For illustrative purposes, we use a PCE that was directly trained by the global Se lifetime (i.e. showing a higher validation error than the surrogate model we use in the paper). For this PCE, we can accurately account for all individual indices, and find a total dummy aerosol Sobol' index of 0.20. This can be compared to the aggregated index that we would have calculated from summing the total indices of the aerosol parameters, 0.23. We do not find a large shift in the total index, and we believe that our approach in the paper is appropriate for summarizing the overall effect of the dummy aerosols, given the surrogate models that we use.

In Figure 5, the main effects are quite low and nowhere near adding up to 1 – was the amount of interaction in this model expected?

We did expect that many non-linear terms involving multiple input parameters could be involved in the Se lifetime, due to the structure of the model. There are many examples that we could provide of interactive effects that could occur. For example if there are low emissions of a certain substance, its rate constant will not matter in the sensitivity analysis. If there are higher emissions of a certain substance, its rate constant could affect the lifetime more, depending on its order of magnitude. Additionally, the dummy aerosol parameters of radius and emissions also would contribute to the variance interactively, as discussed in the paper.

Still with Figure 5, in previous experience seeing interactions that are large and consistent between multiple variables is a sign that the model fit is actually poor. It's not clear because of the way you have carried out LOO and added the interaction terms whether this is indicating poor model fit or whether these are real interactions.

As we showed above, the surrogate model for lifetime does not perfectly capture the model behaviour, however it still shows reasonable accuracy. The large interaction effect is not necessarily a sign of poor model fit, but it could be an intrinsic reason why it is difficult to produce a surrogate model of the lifetime with the quantity of training runs we ran (400, more than the usual rule of thumb of 10 times the number of parameters). More training runs are likely needed to fully capture the interactive behaviour involved in modelling the Se lifetime.

The main effect figures show that there is not much range on the y-axis covered by the central line – it's highlighted by the uncertainty in the remaining parameters. Could you add some information on how much uncertainty there is from using PCE as your surrogate? I would like to see this to show me that most of what you are seeing is not simply a result of the use of a surrogate model.

We have shown above the validation error of our surrogate model, which is not insignificant (0.35). This error could be reduced by running more training runs for the PCE models, however it is not clear how quickly the error will decrease with additional runs. Given that our size of the training set is already quite large and computationally expensive, we had to accept the current accuracy of the surrogate model and proceed with our analysis. In the paper, the sensitivity analysis of the global Se lifetime focuses on the distribution of the simulated lifetime, the ranking of the most important parameters, and visualizing relationships between the lifetime and the input parameters. We do not believe that the main conclusions of our lifetime analysis would change with an expanded training set, even though the Sobol' indices would be refined. The sensitivity analysis of Se deposition would not be affected by these issues, since the LOO and validation errors are, in general, on the order of $10^{-2}$.

**References**

Amouroux, D., Liss, P. S., Tessier, E., Hamren-Larsson, M., and Donard, O. F.: Role of oceans as biogenic sources of selenium, Earth and Planetary Science Letters, 189, 277–283, URL https://doi.org/10.1016/S0012-821X(01)00370-3, 2001.

Blatman, G. and Sudret, B.: An adaptive algorithm to build up sparse polynomial chaos expansions for stochastic finite element analysis, Probabilistic Engineering Mechanics, 25, 183–197, URL https://doi.org/10.1016/j.probengmech.2009.10.003, 2010.

Burkholder, J., Sander, S., Abbatt, J., Barker, J., Huie, R., Kolb, C., Kurylo, M., Orkin, V., Wilmouth, D., and Wine, P.: Chemical Kinetics and Photochemical Data for Use in Atmospheric Studies, Evaluation No. 18, Tech. Rep. 15-10, Jet Propulsion Laboratory, Pasadena, URL https://jpldataeval.jpl.nasa.gov/pdf/JPL_Publication_15-10.pdf, last access: 30 August 2019, 2015.

Lee, K., Hong, S.-B., Lee, J., Chung, J., Hur, S.-D., and Hong, S.: Seasonal variation in the input of atmospheric selenium to northwestern Greenland snow, Science of the Total Environment, 526, 49–57, URL http://doi.org/10.1016/j.scitotenv.2015.04.082, 2015.

Marelli, S. and Sudret, B.: UQLab user manual – Polynomial chaos expansions, Tech. rep., Chair of Risk, Safety & Uncertainty Quantification, ETH Zurich, report # UQLab-V1.3-10, 2019.

Molinaro, A. M., Simon, R., and Pfeiffer, R. M.: Prediction error estimation: a comparison of resampling methods, Bioinformatics, 21, 3301–3307, URL https://doi.org/10.1093/bioinformatics/bti499, 2005.

Sudret, B.: Global sensitivity analysis using polynomial chaos expansions, Reliability Engineering &
System Safety, 93, 964–979, URL `https://doi.org/10.1016/j.ress.2007.04.002`, 2008.

---

## Author Response (AR2)

**Author response to Referee #4 and editor on "Mapping the drivers of uncertainty in atmospheric selenium deposition with global sensitivity analysis"**

We would like to thank Referee #4 and the editor, Dr. Frank Dentener, for having another look at our manuscript and for their comments. We have considered the following comments from the referee and the editor and present our response below:

**Referee #4 comments**

The manuscript adequately addressed the reviewer's comments. However, a shortcoming of the paper is that while the context of the paper is clearly set out in abstract and introduction (0.5-1 billion people suffer from selenium deficit), surprisingly the publication does not discuss what we have learned now from this study that corroborates this number. What was assumed on atmospheric deposition in those health impact estimates, what do we know better now, and how would that change the numbers? A paragraph discussing these issues (referring to future work?) would be beneficial.

**Editor (Dr. Frank Dentener) comments**

Dear author, your revised submission has been reviewed by an additional reviewer. I accept this publication subject to minor revision. The reviewer suggest that the manuscript could potentially benefit from a proper discussion on the potential consequences of this study to improve estimates of Se deficiency health outcomes. If your paper can make a convincing case that those estimates would change substantially using deposition estimates from your study, I could recommend this paper as a highlight item. Alternatively, you can decide to keep the manuscripts as is, but it would still be useful to delineate to include in the discussion, how this study can be used to improve impact studies.

**Our response**

As mentioned in your comments, since Se is an essential element required for health, the ultimate motivation for studying biogeochemical Se cycling is to better understand how this element enters food chains and ecosystems. Agricultural soils stand at the base of the terrestrial food chain and thus play a key role in supplying Se to food crops and animal feed. It has been hypothesized that atmospheric Se could be an important source of Se to soils and thus terrestrial food chains (Winkel et al., 2015); however, until now this could not be proven due to missing knowledge of global atmospheric Se cycling. SOCOL-AER, as the first model of its kind for Se, is now giving us the possibility to quantify atmospheric inputs of Se to terrestrial systems and further assess the pathways of Se in soils and crops, under changing climatic conditions and changes in emissions. For example, in future work the atmospheric model could be coupled to a soil model, such as the machine learning model in Jones et al. (2017), which was previously developed by our group. In terms of atmospheric inputs, the Jones et al. (2017) soil Se model included precipitation as a predictor variable, but a definite improvement would be to use the Se deposition maps as well. Although we cannot directly link atmospheric deposition to nutritional quality and human health effects, the atmospheric model is a crucial step forward in understanding the biogeochemical cycle of Se

and quantifying Se inputs to terrestrial systems, which ultimately will affect nutrition and human health. To summarize this topic, we have added a paragraph at the end of the discussion, P32L28–34:

"The ultimate motivation for studying biogeochemical Se cycling is to better understand how this essential element enters food chains and ecosystems. It has been hypothesized that atmospheric Se could be an important source of Se to soils and thus terrestrial food chains (Winkel et al., 2015); however, until now this could not be proven due to missing knowledge of global atmospheric Se cycling. Using SOCOL-AER, the first model of its kind for Se, we can quantify the atmospheric inputs of Se to terrestrial systems and investigate how these inputs are impacted by different climate conditions and anthropogenic activities. Although human health effects cannot be directly inferred from atmospheric deposition maps, SOCOL-AER will be coupled to soil, plant, and health system models in future work to accurately predict Se deficiency risks. "

We have also added a sentence concerning these future plans in the conclusion, P33L12–13:

"In future work, deposition maps from SOCOL-AER can be linked to soil, plant, and health system models to identify the regions at risk of Se deficiency in current times and the future."

In addition, we have corrected several typos that had been included in Table 3 for the parameter bounds of the chemical rate constants.

**References**

[revised manuscript text omitted]